# OPENS2V-NEXUS: A Detailed Benchmark and Million-Scale Dataset for Subject-to-Video Generation

**Shenghai Yuan**[1,4,*], **Xianyi He**[1,4,*], **Yufan Deng**[1], **Yang Ye**[1,4], **Jinfa Huang**[3],
**Bin Lin**[1,4], **Jiebo Luo**[3], **Li Yuan**[1,2,†]

∗ Equal Contributors, † Corresponding Authors

[1] Peking University, Shenzhen Graduate School, [2] Peng Cheng Laboratory,

[3] University of Rochester, [4] Rabbitpre AI

{yuanshenghai@stu, yuanli-ece@}.pku.edu.cn

## Abstract

Subject-to-Video (S2V) generation aims to create videos that faithfully incorporate reference content, providing enhanced flexibility in the production of videos. To establish the infrastructure for S2V generation, we propose **OPENS2V-NEXUS**, consisting of (i) *OpenS2V-Eval*, a fine-grained benchmark, and (ii) *OpenS2V-5M*, a million-scale dataset. In contrast to existing S2V benchmarks inherited from VBench [38] that focus on global and coarse-grained assessment of generated videos, *OpenS2V-Eval* focuses on the model's ability to generate subject-consistent videos with natural subject appearance and identity fidelity. For these purposes, *OpenS2V-Eval* introduces 180 prompts from seven major categories of S2V, which incorporate both real and synthetic test data. Furthermore, to accurately align human preferences with S2V benchmarks, we propose three automatic metrics, NexusScore, NaturalScore, and GmeScore, to separately quantify subject consistency, naturalness, and text relevance in generated videos. Building on this, we conduct a comprehensive evaluation of 18 representative S2V models, highlighting their strengths and weaknesses across different content. Moreover, we create the first open-source large-scale S2V generation dataset *OpenS2V-5M*, which consists of five million high-quality 720P subject-text-video triples. Specifically, we ensure subject-information diversity in our dataset by (1) segmenting subjects and building pairing information via cross-video associations and (2) prompting GPT-Image on raw frames to synthesize multi-view representations. Through **OPENS2V-NEXUS**, we deliver a robust infrastructure to accelerate future S2V generation research. [1]

## 1 Introduction

With the advancement of video foundational models [52, 92, 62, 130, 43, 73, 89, 109, 115], Subject-to-Video (S2V) generation has attracted increasing attention, enabling the generation of videos centered on reference subjects. Previous tuning-based methods [72, 32, 68, 25] require fine-tuning for each sample during inference, which is time-consuming. Recently, several open-source S2V models [129, 100, 22], including ConsisID [119], Phantom [58], and VACE [42], as well as closed-source models [46, 5, 45, 90, 18], have demonstrated the ability to perform tuning-free S2V generation.

Although these methods demonstrate promising results, there remains a shortage of benchmarks for objectively evaluating the strengths and limitations of S2V models. As shown in Table 1, existing

---

[1]The source data and code are publicly available on https://pku-yuangroup.github.io/OpenS2V-Nexus.

39th Conference on Neural Information Processing Systems (NeurIPS 2025) Track on Datasets and Benchmarks.

Table 1: **Comparison of the Characteristics of our OpenS2V-Eval with existing Benchmarks.** Most of them focus on T2V and neglect the evaluation of subject naturalness. _ means suboptimal.

| Benchmark | # Type | Visual Quality | Text Relevance | Motion Quality | Subject Consistency | Subject Naturalness |
|---|---|---|---|---|---|---|
| Make-a-Video-Eval [84] | Text-to-Video | ✓ | ✓ | ✗ | ✗ | ✗ |
| FETV [61] | Text-to-Video | ✓ | ✓ | ✓ | ✗ | ✗ |
| T2VScore [104] | Text-to-Video | ✓ | ✓ | ✓ | ✗ | ✗ |
| EvalCrafter [60] | Text-to-Video | ✓ | ✓ | ✓ | ✗ | ✗ |
| VBench [38] | Text-to-Video | ✓ | ✓ | ✓ | ✗ | ✗ |
| VBench++ [39] | Text-to-Video | ✓ | ✓ | ✓ | ✗ | ✗ |
| ChronoMagic-Bench [121] | Text-to-Video | ✓ | ✓ | ✓ | ✗ | ✗ |
| ConsisID-Bench [119] | Subject-to-Video | ✓ | ✓ | ✓ | _✓_ | ✗ |
| Alchemist-Bench [13] | Subject-to-Video | ✓ | ✓ | ✓ | _✓_ | ✗ |
| A2 Bench [22] | Subject-to-Video | ✓ | ✓ | ✓ | _✓_ | ✗ |
| VACE-Bench [42] | Subject-to-Video | ✓ | ✓ | ✓ | _✓_ | ✗ |
| **OpenS2V-Eval** | Subject-to-Video | ✓ | ✓ | ✓ | ✓ | ✓ |

video generation benchmarks predominantly focus on text-to-video tasks, with prominent examples including VBench [39] and ChronoMagic-Bench [121]. While ConsisID-Bench [119] is applicable to S2V, it is restricted to assessing facial consistency. Alchemist-Bench [13], VACE-Benchmark [42], and A2 Bench [22] support the evaluation of open-domain S2V; however, their evaluation are primarily global and coarse-grained. For example, they neglect to assess the naturalness of subjects. Furthermore, the latter two benchmarks [42, 22] inherit their subject consistency metrics from VBench [39], which calculates similarity directly between uncropped video frames and reference images—an approach that unavoidably introduces background noise and reduces accuracy.

Subject-to-Video (S2V) models currently face three major challenges: **(1) Poor generalization**: These models often perform poorly when encountering subject categories not seen during training [42, 119]. For instance, a model trained exclusively on Western subjects typically performs worse when generating Asian subjects; **(2) Copy-paste issue**: The model tends to directly transfer the pose, lighting, and contours from the reference image to the video, resulting in unnatural outcomes [22]; **(3) Inadequate human fidelity**: Current models often struggle to preserve human identity as effectively as they do non-human entities [58]. An effective benchmark should be able to identify these issues. However, even when the generated subject appears unnatural or when the fidelity is low, existing benchmarks [42, 22, 127, 116] still yield high scores, hindering progress in the field.

To address this challenge, we introduce OpenS2V-Eval, the first comprehensive subject-to-video benchmark in the field. Specifically, we define seven categories: ① single-face-to-video, ② single-body-to-video, ③ single-entity-to-video, ④ multi-face-to-video, ⑤ multi-body-to-video, ⑥ multi-entity-to-video, and ⑦ human-entity-to-video, as in Figure 1. For each category, we design 30 test samples with rich visual content, which assess the model's generalization ability across different subjects. To address the limited robustness of existing automatic metrics, we first develop NexusScore, which combines an image-prompt detection model [15] and a multimodal retrieval model [125] to accurately evaluate subject consistency. Next, we introduce NaturalScore, a GPT-based metric designed to bridge the gap in evaluating subject naturalness. Finally, we propose GmeScore, based on MLLM [125], which provides a more precise assessment of text relevance compared to conventional CLIPScore [76]. Using OpenS2V-Eval, we conduct both qualitative and quantitative evaluations of nearly all open-source and closed-source S2V models, offering valuable insights for model selection.

Furthermore, when the community attempts to extend foundational models to downstream tasks, existing datasets are limited in their support for complex tasks [8, 33, 72, 85, 34, 86, 64], as shown in Table 2. To address this limitation, we propose OpenS2V-5M, the first million-scale dataset specifically designed for subject-to-video, which is also applicable to text-to-video [81, 26, 103]. Unlike previous methods [119, 42, 22, 13, 58] that rely solely on regular subject-text-video triples—where subject images are segmented from training frames, potentially causing the model to learn shortcuts rather than intrinsic knowledge—we enrich it with Nexus Data, through (1) building pairing information via cross-video associations and (2) prompting GPT-Image-1 [1] on raw frames to synthesize multi-view representations, to address the three core challenges mentioned above at the data level.

The contributions of this work are as follows:

**i) New S2V Benchmark.** We introduce *OpenS2V-Eval* for comprehensive evaluation of S2V models and propose three new automatic metrics aligned with human perception.

**ii) New Insights for S2V Model Selection.** Our evaluations using *OpenS2V-Eval* provide crucial insights into the strengths and weaknesses of various subject-to-video generation models.

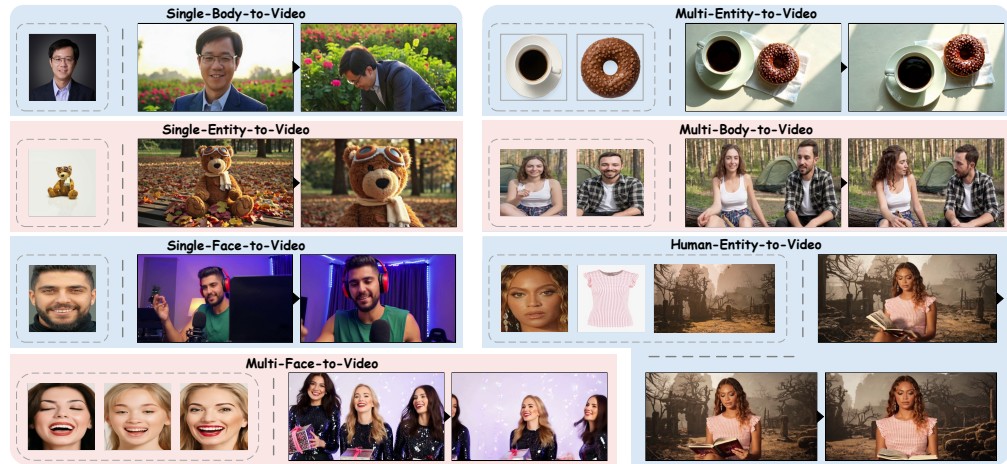

Figure 1: **Example of Seven Categories from OpenS2V-Eval.** These categories fully encompass the subject-to-video tasks, allowing comprehensive evaluation. Videos are generated by Kling [45].

Table 2: **Comparison of the Statistics of OpenS2V-5M with existing Video Generation Datasets.** Most of them are inadequate for extending foundational models to subject-to-video generation task.

| Dataset | # Type | Resolution | Video Clips | Average Length (s) | Video Duration (h) |
|---|---|---|---|---|---|
| MSRVTT [110] | Text-to-Video | 240P | 10K | 14.4 | 40 |
| WebVid-10M [4] | Text-to-Video | 360P | 10M | 18.7 | 52K |
| InternVid [98] | Text-to-Video | 720p | 234M | 11.7 | 760K |
| HD-VG-130M [97] | Text-to-Video | 720p | 130M | 4.9 | 178K |
| Panda-70M [12] | Text-to-Video | 720P | 70M | 8.6 | 167K |
| OpenVid-1M [70] | Text-to-Video | 512P | 1M | 7.2 | 2K |
| Koala-36M [94] | Text-to-Video | 720P | 36M | 17.2 | 172K |
| ChronoMagic-Pro [121] | Text-to-Video | 720p | 460K | 234.8 | 30K |
| OpenHumanVid [47] | Text-to-Video | 720P | 52.3M | 4.9 | 70K |
| **OpenS2V-5M** | Subject-to-Video | 720P | 5.4M | 6.6 | 10K |

**iii) Large-Scale S2V Dataset.** We create *OpenS2V-5M*, a dataset with 5.1M high-quality regular data and 0.35M Nexus Data, the latter is expected to address the three core challenges of subject-to-video.

## 2  Related Work

**Automatic Metrics for Subject-to-Video Generation.**     Existing video generation benchmarks typically focus on text-to-video tasks [44, 105, 112, 99, 20, 30]. Notable examples include MSR-VTT [110] and Make-a-Video-Eval [84], which are pioneering benchmarks for video generation evaluation. Later, VBench [38, 39, 127] and EvalCrafter [60] consider multiple evaluation dimensions, providing a more comprehensive benchmark by considering additional mode-specific factors. ConsisID-Bench [119] represents an early work for S2V, but is limited to human domain. Although recent benchmarks, such as A2 Bench [22] and VACE-Benchmark [42], are applicable to open-domain S2V tasks, they rely on VBench [38] metrics to calculate subject consistency without being specifically tailored for S2V. Therefore, we develop the first comprehensive subject-to-video benchmark, which includes 180 balanced test pairs. Furthermore, we introduce NexusScore, NaturalScore, GmeScore to accurately measure subject consistency, naturalness, and text relevance, thereby addressing this gap in the field.

**Datasets for Subject-to-Video Generation.**     Large-scale, high-quality video datasets [4, 98, 97, 70, 96] are essential to emerging DiT-based generation model [124, 82, 57, 7, 21, 57, 63, 117, 54, 128]. For instance, newly released Panda-70M [12], Koala-36M [94], and ChronoMagic-Pro [121] feature millions of high-resolution video-text pairs, which have substantially contributed to the progress of the field. However, when the community seeks to extend the foundational model to downstream tasks, existing open-source datasets are inadequate for subject-to-video [18, 58]. Moreover, we identify a significant issue, whether the model is closed-source [46, 5, 45] or open-source: they all suffer the

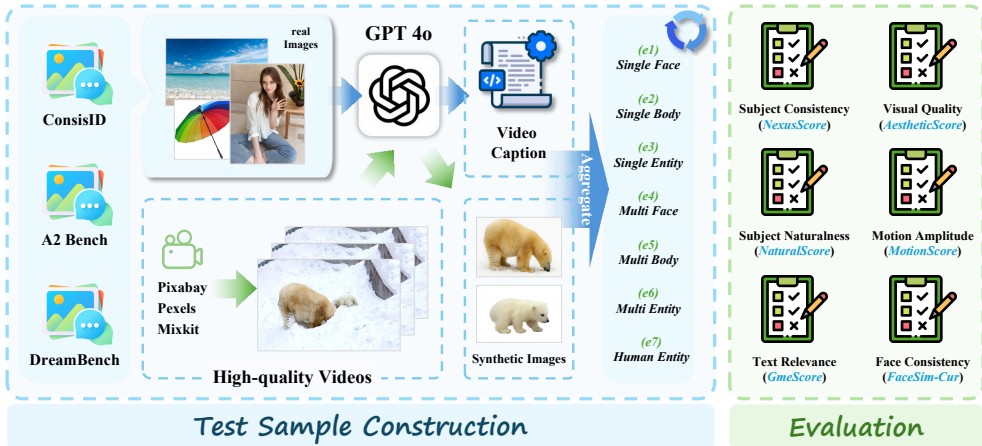

Figure 2: **The Pipeline of Constructing OpenS2V-Eval.** (Left) Our benchmark includes not only real subject images but also synthetic images constructed through GPT-Image-1 [1], allowing for a more comprehensive evaluation. (Right) The metrics are tailored for subject-to-video generation, evaluating not only S2V characteristics (e.g., consistency) but also basic video elements (e.g., motion).

three core issues of subject-to-video mentioned above. To address this gap, we introduce the first million-scale subject-to-video dataset, named OpenS2V-5M. In addition to extracting subject images from segmented training frames, we further propose constructing subject images through building pairing information and synthesis using GPT-Image-1 [1], thereby empowering the community.

## 3 OpenS2V-Eval

### 3.1 Prompt Construction

To comprehensively evaluate the capabilities of subject-to-video models [18, 58, 23], the designed text prompts must encompass a wide range of categories, and the corresponding reference images must meet high-quality standards. Consequently, to construct a benchmark for subject-to-video that incorporates diverse visual concepts, we divide this task into seven categories: ① single-face-to-video, ② single-body-to-video, ③ single-entity-to-video, ④ multi-face-to-video, ⑤ multi-body-to-video, ⑥ multi-entity-to-video, and ⑦ human-entity-to-video. Based on this, we collect 50 and 24 subject-text pairs from ConsisID [119] and A2 Bench [22], respectively, for constructing ①, ②, and ⑥. Additionally, we gather 30 reference images from DreamBench [74] and utilized GPT-4o [1] to generate captions for building ③. Subsequently, we source high-quality videos from copyright-free websites, employ GPT-Image-1 [1] to extract subject images from the videos, and use GPT-4o to caption the videos, thereby obtaining the remaining subject-text pairs. Collection for each sample is performed manually to ensure benchmark quality. Unlike prior benchmark [13, 42] that relied solely on real images, the inclusion of synthetic samples enhances the diversity and precision of evaluation.

### 3.2 Benchmark Statistics

We collect 180 high-quality subject-text pairs, consisting of 80 real and 100 synthetic samples. Except for ④ and ⑤, which each contain 15 samples, all other categories include 30 samples. The data statistics are shown in Figure 3. As illustrated in (c) and (d), the seven major categories of the S2V task encompass a broad range of testing scenarios, including various objects, backgrounds and actions. Additionally, terms associated with humans, such as "woman" and "man," make up a significant proportion, allowing for a comprehensive evaluation of existing methods' ability to preserve human identity—an especially challenging aspect of the S2V task. Furthermore, since some methods prefer long captions [42] while others prefer short ones [58], we ensure that the text prompts vary in length, as shown in (b). We also assess the aesthetic scores of the collected reference images, with the results showing that most score above 5, indicating high quality. Moreover, we retain some lower-quality images to preserve the diversity of evaluation. Due to the limitations of existing S2V models [45, 18, 46], we restrict the number of subject images for each sample to no more than three.

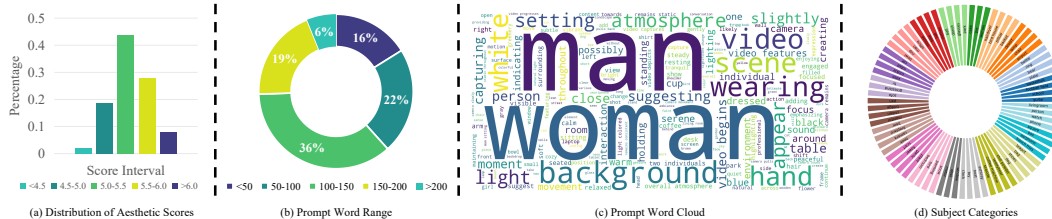

(a) Distribution of Aesthetic Scores    (b) Prompt Word Range    (c) Prompt Word Cloud    (d) Subject Categories

Figure 3: **Statistics in OpenS2V-Eval.** The benchmark covers diverse categories and prompt words, with subject images displaying high aesthetics, thus enabling a thorough evaluation.

## 3.3 New Automatic Metrics

As previously mentioned, existing S2V benchmarks are usually adapted from T2V rather than being specifically tailored. For subject-to-video, it is crucial to evaluate not only global aspects such as visual quality and motion but also subject consistency and naturalness in the synthesized output.

**NexusScore**    To calculate subject consistency, prior studies [42, 58, 22, 38, 39] directly compute the similarity between uncropped video frames and reference images in the DINO [122] or CLIP [76] space. However, this method introduces background noise, and the feature space has been proven to be unreasonable [104, 61, 121]; please refer to the Appendix B.1 for more details. To address this issue, we introduce the NexusScore $S_{\text{Nexus}}$, which utilizes the image-prompt detection model $\mathcal{M}_{\text{detect}}$ [15] and the multimodal retrieval model $\mathcal{M}_{\text{retrieve}}$ [125]. Specifically, both the reference images $\{R_i\}_{i=1}^I$ and video frames $\{I_t\}_{t=1}^T$ are firstly fed into the $\mathcal{M}_{\text{detect}}$, which identifies the relevant target in each frame and generates the corresponding bounding box $B_{i,t}$ that encloses the target:

$$B_{i,t} = \mathcal{M}_{\text{detect}}(R_i, I_t), \tag{1}$$

To improve the accuracy of the bounding box, for each subject, we crop the region $B_{i,t}$ to get the cropped reference image $C_{i,t}$. Then, we compute the similarity between the cropped reference image $C_{i,t}$ and the corresponding target entity name $E_{i,t}$ in the unified text-image feature space. This similarity is denoted as $s$, and it is computed using the multimodal retrieval model $\mathcal{M}_{\text{retrieve}}$:

$$s_{i,t} = \mathcal{M}_{\text{retrieve}}(C_{i,t}, E_{i,t}), \tag{2}$$

If bbox $B_{i,t}$ confidence $c_{i,t}$ and $s_{i,t}$ exceeds a predefined threshold $\alpha$ and $\beta$, we proceed to the next stage. Finally, the similarity between $C_{i,t}$ and $R_i$ is evaluated in the image feature space, yielding:

$$S_{\text{Nexus}} = \frac{1}{I \times T'} \sum_{i=0}^{I} \sum_{t=0}^{T'} \mathcal{M}_{\text{retrieve}}(C_{i,t}, R_i), \quad \text{where} \quad c_{i,t} > \alpha \quad \text{and} \quad s_{i,t} > \beta \tag{3}$$

where $T'$ means the total number of frames in which an object is detected. Appendix D.4 for details.

**NaturalScore**    Unlike existing subject-to-video benchmarks [119, 22, 42, 58] that focus exclusively on subject consistency, we additionally evaluate whether the generated subject appears natural, i.e., whether it conforms to physical laws. This is due to the prevalent "copy-paste" issue in current S2V methods, where the model blindly copies the reference image onto the generated scene, resulting in high consistency scores even when the output fails to align with typical human perception.

To address this issue, a straightforward solution is to employ the AIGC anomaly detection model [111, 48, 69]. However, we found that the accuracy of open-source models is suboptimal. An alternative approach is to utilize open-source multimodal large language models [3, 53, 88] for video scoring. However, these models exhibit poor instruction-following performance and are prone to significant hallucinations. For a more details, please refer to Appendix B.2. As a result, we use GPT-4o [1] to simulate human evaluators, which provides superior accuracy and flexibility. Specifically, we subtly design a five-point evaluation criterion based on common sense and physical laws, denoted as $C = \{c_1, c_2, c_3, c_4, c_5\}$, where each $c_i$ represents a score corresponding to a specific evaluation level. For each video, we uniformly sample $T$ frames, denoted as $\{I_t\}_{t=1}^T$. These frames are then input into GPT-4o $\mathcal{M}_{\text{GPT}}$, which assigns a score $s_t$ and provides reasoning based on the five-point scale. The final score $S_{\text{Natural}}$ is computed as the average of the scores assigned to all $T$ frames:

$$S_{\text{Natural}} = \frac{1}{T} \sum_{t=1}^{T} \mathcal{M}_{\text{GPT}}(I_t) \tag{4}$$

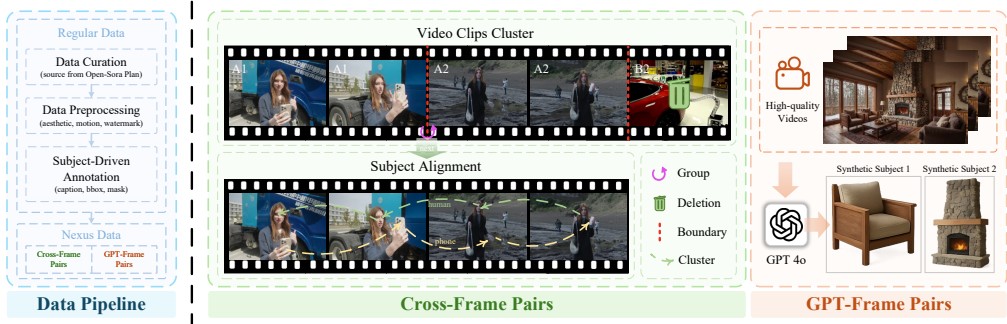

Figure 4: **The Pipeline of Constructing OpenS2V-5M.** First, we filter low-quality videos based on scores such as aesthetics and motion, then utilize GroundingDino [59] and SAM2.1 [79] to extract subject images and get Regular Data. Subsequently, we create Nexus Data through cross-video association and GPT-Image-1 [1] to address the three core issues encountered by S2V models.

**GmeScore**   Existing methods commonly calculate text relevance using CLIP [76] or BLIP [123]. However, several studies [61, 121, 104] have identified inherent flaws in these models' feature spaces, resulting in inaccurate scores. Additionally, their text encoders are limited to 77 tokens, which makes them unsuitable for the long text prompts preferred by current DiT-based video generation models [62, 82, 113, 92]. In light of this, we opt to utilize GME [125], a model fine-tuned on Qwen2-VL [93], which naturally accommodates text prompts of varying lengths and yields more reliable scores.

## 4   OpenS2V-5M

### 4.1   Data Construction

**Subject-Driven Processing.**   As noted previously, existing large-scale video generation datasets typically consist only of text and video [121, 12, 94, 47], limiting their applicability for developing complex subject-to-video tasks. To overcome this limitation, we develop the first large-scale subject-to-video dataset, with raw videos sourced from Open-Sora Plan [52]. Given that the metadata includes video captions, we initially select videos featuring human, as these tend to contain a larger number of subjects. Next, we filter out low-quality video based on aesthetic [16], motion [6], and technical scores [102], resulting in 5,437,544 video clips. Building on this, and following the ConsisID data pipeline [119], we utilize Grounding DINO [59] and SAM2.1 [79] to extract subjects from each video, yielding regular data suitable for subject-to-video tasks. Finally, to ensure data quality, we assign aesthetic score and GmeScore to the reference images using the aesthetic [16] and multimodal retrieval models [125], enabling users to adjust thresholds to balance data quantity and quality.

**Generalized Nexus Construction.**   Existing S2V methods primarily rely on regular data, where the extracted subject often shares the same view as the one in the training frames and may be incomplete, leading to the three core challenges discussed in Section 1. This limitation arises due to the extraction of the reference image directly from the ground truth video, leading the model to take shortcuts by copying the reference image onto the generated video instead of learning the underlying knowledge, reducing generalization. To overcome this, we introduce Nexus Data, including GPT-Frame Pairs and Cross-Frame Pairs. Comparison between regular data and Nexus Data is shown in Figure 5.

For GPT-Frame Pairs: let $I_0$ represent the first frame of a given video, and let $K = \{k_1, k_2, \ldots, k_n\}$ be a set of keywords associated with the subject of the video. We input $I_0$ and $K$ into GPT-Image-1 [1] $\mathcal{M}_{\text{GPT}}$, which then generates a complete image $I_{\text{gen}}$ of the corresponding subject, forming the pair $\langle I_0, I_{\text{gen}} \rangle$, which we refer to as *GPT-Frame Pairs*. Due to the powerful generative capabilities of GPT-Image-1, it can reconstruct incomplete subjects and generate consistent content from multiple perspectives, ensuring alignment with our data requirements. This relationship can be formalized as:

$$I_{\text{gen}} = \mathcal{M}_{\text{GPT}}(I_0, K) \tag{5}$$

For Cross-Frame Pairs: since clips are split from long videos, where there exists an inherent temporal and semantic correlation between clips [129]. To capture this, we aggregate clips from the same long video, denoted as $C = \{C_1, C_2, \ldots, C_m\}$, where each $C_i$ corresponds to a different segment of the

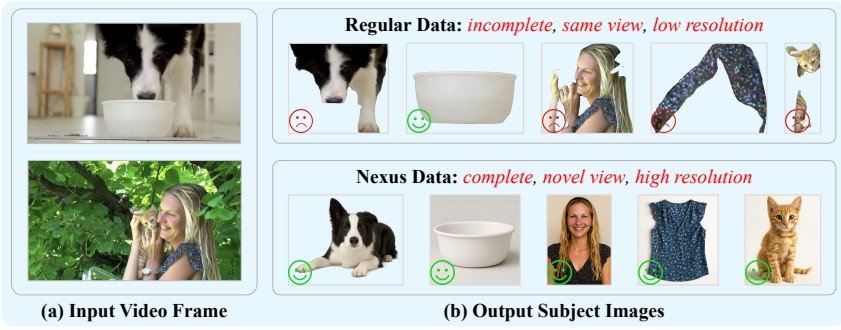

**(a) Input Video Frame**    **(b) Output Subject Images**

Figure 5: **Comparison between Regular Data and Nexus Data.** The latter is of higher quality.

video. The similarity between subjects across these clips is computed using a multimodal retrieval model [125] $\mathcal{M}_{\text{retrieval}}$, which computes the similarity score $S(C_{ij}, C_{kl})$ for any pair of clips $C_{ij}$ and $C_{kl}$, where $i \neq k$ represents different segments of the video, and $j$ and $l$ represent different subjects:

$$S(C_{ij}, C_{kl}) = \text{sim}(\mathcal{M}_{\text{retrieval}}(C_{ij}), \mathcal{M}_{\text{retrieval}}(C_{kl})) \qquad (6)$$

where $\text{sim}(\cdot, \cdot)$ means computing the similarity. This process enable the formation of *Cross-Frame Pairs* $\langle C_{ij}, C_{kl} \rangle$. Finally, we assign aesthetic score [16] and GmeScore to each sample.

### 4.2 Dataset Statistics

OpenS2V-5M is the first open-source million-scale subject-to-video dataset. It includes 5.1M regular data, commonly used in existing methods [42, 22, 58], as well as 0.35M Nexus Data, generated through GPT-Image-1 [1] and cross-video associations. This dataset is anticipated to address the three core challenges faced by S2V models. Detailed statistics can be found in the Appendix C.2.

## 5 Experiments

### 5.1 Evaluation Setups

**Evaluation Baseline.**    We evaluate almost all S2V models, including four closed-source and fourteen open ones, including models that support all type of subject (e.g., Vidu [5], Pika [46], Kling [45], VACE [42], Phantom [58], SkyReels-A2 [22], and HunyuanCustom [35]), as well as models that only support human identity (e.g., Hailuo [90], ConsisID [119], Concat-ID [129], FantasyID [126], EchoVideo [100], VideoMaker [107], and ID-Animator [31]).

**Application Scope.**    OpenS2V-Eval presents an automated scoring method for evaluating subject consistency, subject naturalness, and text relevance. By incorporating existing metrics for visual quality, motion quality, and face similarity (e.g., Aesthetic Score [16], Motion Amplitude [6], Motion Smoothness [55], and FaceSim-Cur [119]), it facilitates an evaluation of the S2V model across six dimensions. Furthermore, human evaluation can be utilized to provide a more precise assessment.

**Implementation Details.**    Closed-source S2V models can only perform manually through their interfaces, and the inference speed of open-source models is relatively slow (e.g., VACE-14B [42] requires over 50 mins to get a $81 \times 720 \times 1280$ video on a single Nvidia A100). Therefore, for each baseline, we only generate a video for each test sample in OpenS2V-Eval. We then evaluate all generated videos using the six aforementioned automated metrics. All inference settings follow the official implementation, with the seed fixed at 42. Further details are provided in the Appendix D.

### 5.2 Comprehensive Analysis

**Quantitative Evaluation.**    We first present a comprehensive qualitative evaluation of different methods, with results displayed in Table 3, 4, and 5. All models are capable of generating videos with high visual quality and text relevance. For open-domain S2V, closed-source models generally outperform their open-source counterparts. Among these, Pika [46] achieves the highest GmeScore, indicating that the generated videos are better aligned with the provided instructions. Kling [45], on

Table 3: **Quantitative Comparison among Different Methods for the Open-Domain Subject-to-Video task.** Total score is the normalized weighted sum of other scores. "↑" higher is better.

| Method | Venue | Total Score↑ | Aesthetics↑ | Motion-A↑ | Motion-S↑ | FaceSim↑ | GmeScore↑ | NexusScore↑ | NaturalScore↑ |
|---|---|---|---|---|---|---|---|---|---|
| Vidu2.0 [5] | Closed-Source | 51.95% | 41.48% | 13.52% | 90.45% | 35.11% | 67.57% | 43.37% | 65.88% |
| Pika2.1 [46] | Closed-Source | 51.88% | 46.88% | 24.71% | 87.06% | 30.38% | 69.19% | 45.40% | 63.32% |
| Kling1.6 [45] | Closed-Source | 56.23% | 44.59% | **41.60%** | 86.93% | 40.10% | 66.20% | **45.89%** | 74.59% |
| VACE-P1.3B [42] | Open-Source | 48.98% | 47.34% | 12.03% | 96.80% | 16.59% | **71.38%** | 40.19% | 64.31% |
| VACE-1.3B [42] | Open-Source | 49.89% | **48.24%** | 18.83% | **97.20%** | 20.57% | 71.26% | 37.91% | 65.46% |
| VACE-14B [42] | Open-Source | 57.55% | 47.21% | 15.02% | 94.97% | 55.09% | 67.27% | 44.08% | 67.04% |
| Phantom-1.3B [58] | Open-Source | 54.89% | 46.67% | 14.29% | 93.30% | 48.56% | 69.43% | 42.48% | 62.50% |
| Phantom-14B [58] | Open-Source | 56.77% | 46.39% | 33.42% | 96.31% | 51.46% | 70.65% | 37.43% | 69.35% |
| SkyReels-A2-P14B [22] | Open-Source | 52.25% | 39.41% | 25.60% | 87.93% | 45.95% | 64.54% | 43.75% | 60.32% |
| MAGREF-480P [19] | Open-Source | 52.51% | 45.02% | 21.81% | 93.17% | 30.83% | 70.47% | 43.04% | 66.90% |

Table 4: **Quantitative Comparison among Different Methods for the Human-Domain Subject-to-Video task.** Total score is the normalized weighted sum of other scores. "↑" higher is better.

| Method | Venue | Domain | Total Score↑ | Aesthetics↑ | Motion-A↑ | Motion-S↑ | FaceSim↑ | GmeScore↑ | NaturalScore↑ |
|---|---|---|---|---|---|---|---|---|---|
| Vidu2.0 [5] | Closed-Source | Open-Domain | 57.70% | 47.33% | 14.54% | 91.31% | 38.50% | 70.43% | 67.78% |
| Pika2.1 [46] | Closed-Source | Open-Domain | 56.84% | 52.39% | 28.77% | 85.29% | 29.42% | **75.03%** | 67.53% |
| Kling1.6 [45] | Closed-Source | Open-Domain | 60.19% | 50.94% | 50.02% | 84.75% | 41.02% | 67.79% | **71.55%** |
| VACE-P1.3B [42] | Open-Source | Open-Domain | 53.97% | 51.91% | 8.78% | 95.80% | 19.98% | 73.27% | 65.83% |
| VACE-1.3B [42] | Open-Source | Open-Domain | 54.90% | 53.18% | 16.87% | 95.84% | 22.29% | 73.61% | 65.28% |
| VACE-14B [42] | Open-Source | Open-Domain | 65.78% | 52.78% | 11.76% | 94.96% | **64.65%** | 69.53% | 69.31% |
| Phantom-1.3B [58] | Open-Source | Open-Domain | 60.00% | 50.80% | 14.09% | 92.02% | 46.29% | 72.17% | 65.83% |
| Phantom-14B [58] | Open-Source | Open-Domain | 64.22% | 49.14% | 41.24% | 94.81% | 55.04% | 72.55% | 69.86% |
| SkyReels-A2-P14B [22] | Open-Source | Open-Domain | 56.43% | 39.89% | 31.49% | 80.19% | 55.01% | 63.63% | 59.31% |
| HunyuanCustom [35] | Open-Source | Open-Domain | 61.22% | 49.67% | 15.13% | 84.73% | 62.25% | 69.78% | 60.56% |
| MAGREF-480P [19] | Open-Source | Open-Domain | 57.72% | 51.2% | 14.76% | 90.26% | 32.87% | 70.88% | 70.28% |
| Hailuo [90] | Closed-Source | Human-Domain | 65.26% | **52.75%** | 31.80% | **99.10%** | 57.69% | 71.42% | 69.20% |
| ConsisID [119] | Open-Source | Human-Domain | 54.19% | 41.77% | 37.99% | 79.83% | 43.19% | 72.03% | 55.83% |
| Concat-ID-CogVideoX [129] | Open-Source | Human-Domain | 55.89% | 44.13% | 31.07% | 81.90% | 43.87% | 73.67% | 58.75% |
| Concat-ID-Wan-AdaLN [129] | Open-Source | Human-Domain | 59.85% | 43.13% | 17.19% | 85.86% | 50.05% | 71.90% | 68.47% |
| FantasyID [126] | Open-Source | Human-Domain | 54.33% | 45.60% | 23.41% | 85.44% | 32.48% | 72.68% | 62.36% |
| EchoVideo [100] | Open-Source | Human-Domain | 56.36% | 39.93% | 35.58% | 77.96% | 48.65% | 68.40% | 62.22% |
| VideoMaker [107] | Open-Source | Human-Domain | 54.23% | 31.76% | **50.09%** | 77.5% | 76.45% | 45.28% | 47.08% |
| ID-Animator [31] | Open-Source | Human-Domain | 49.75% | 42.03% | 33.54% | 94.69% | 31.56% | 52.91% | 56.11% |
| Ours † | - | Human-Domain | 58.00% | 41.30% | 20.83% | 84.32% | 47.64% | 72.12% | 65.42% |
| Ours ‡ | - | Human-Domain | 59.23% (+1.23%) | 41.86% (+0.56%) | 22.77% (+1.94%) | 86.03% (+1.71%) | 49.51% (+1.87%) | 72.35% (+0.23%) | 66.80% (+1.38%) |

the other hand, produces videos with higher fidelity and realism, securing the highest NexusScore and NaturalScore. While SkyReels-A2 [22] holds the high NexusScore among open-source models, its relatively low NaturalScore suggests the presence of a copy-paste issue. VACE-1.3B and VACE-14B [42] achieve superior generation quality across the board compared to the VACE-P1.3B [42] by scaling both the parameter size and the dataset. In the human-domain S2V task, proprietary models outperform open-domain models in terms of preserving human identity, particularly Hailuo [90], which achieves the highest Total Score of 60.20%. Furthermore, NaturalScore reveals that open-source models such as ConsisID [119] and Concat-ID [129], despite having relatively strong FaceSim, suffer from significant copy-paste issues. In contrast, EchoVideo [100] achieves the highest score among the open-source human-domain models. Since HunyuanCustom [35] only released the single-subject version as open source, we additionally provide results for the single-domain scenario, as presented in Table 5. Notably, although HunyuanCustom [35] achieves high subject fidelity, its generated styles tend to exhibit artificial characteristics, resulting in less realistic outputs.

**Qualitative Evaluation.** Next, we randomly select three test data for qualitative analysis, as shown in Figures 6, 7, and 8. Overall, closed-source models exhibit a clear advantage in terms of overall capability (e.g., Kling [45]). Open-source models, represented by Phantom [58] and VACE [42], are closing this gap; however, both models share the following three common issues: **(1) Poor generalization:** Fidelity is low for certain subjects. For instance, in case 2 of Figure 6, Kling [45] generates an incorrect playground background, while VACE [42], Phantom [58], and SkyReels-A2 [22] produce low-fidelity humans and birds; **(2) Copy-paste issues:** In Figure 7, SkyReels-A2 [22] and VACE [42] incorrectly replicate the expression, lighting, or pose from the reference image into the generated video, resulting in unnatural output; **(3) Inadequate human fidelity:** In case 2 of Figure 6, only Kling [45] maintains human identity in the first half of the video, while the other models lose significant facial details throughout the video. Figure 7 shows that all models fail to accurately render the profile of the individual. Additionally, we observe that (1) As the number of reference images increases, fidelity gradually decreases; (2) the initial frames may blurry or directly copied; (3) fidelity gradually declines over time. For more details, please refer to the Appendix B.4.

**Human Preference.** Then, we validate the effectiveness of metrics through manual cross-validation. Sixty generated videos corresponding to the prompts are randomly selected, and 173 participants are invited to vote, yielding evaluation results. To improve user satisfaction, we employ a binary classification questionnaire format. Figure 9(a) illustrates the correlation between the automatic metrics and human perception. It is evident that the three proposed metrics—Nexus Score, NaturalScore,

Table 5: **Quantitative Comparison among Different Methods for the Single-Domain Subject-to-Video task.** Total score is the normalized weighted sum of other scores. "↑" higher is better.

| Method | Venue | Total Score↑ | Aesthetics↑ | Motion-A↑ | Motion-S↑ | FaceSim↑ | GmeScore↑ | NexusScore↑ | NaturalScore↑ |
|---|---|---|---|---|---|---|---|---|---|
| Vidu2.0 [5] | Closed-Source | 52.90% | 43.32% | 17.52% | 91.88% | 36.19% | 66.96% | 44.84% | 66.11% |
| Pika2.1 [46] | Closed-Source | 53.12% | 47.43% | 26.32% | 86.07% | 32.33% | 69.84% | 47.35% | 64.68% |
| Kling1.6 [45] | Closed-Source | 56.67% | 45.97% | **47.17%** | 85.76% | 39.27% | 65.36% | 49.30% | **73.63%** |
| VACE-P1.3B [42] | Open-Source | 49.20% | 48.93% | 11.91% | **95.68%** | 18.04% | 70.78% | 36.24% | 66.85% |
| VACE-1.3B [42] | Open-Source | 51.13% | **49.41%** | 22.51% | 95.42% | 22.37% | **70.87%** | 38.34% | 68.33% |
| VACE-14B [42] | Open-Source | **61.75%** | 48.94% | 19.69% | 93.16% | **64.65%** | 65.86% | 50.82% | 70.56% |
| Phantom-1.3B [58] | Open-Source | 54.50% | 49.00% | 16.38% | 93.70% | 44.03% | 69.54% | 37.72% | 66.76% |
| Phantom-14B [58] | Open-Source | 57.02% | 47.46% | 41.55% | 94.86% | 51.82% | 70.07% | 35.30% | 71.11% |
| SkyReels-A2-P14B [22] | Open-Source | 55.06% | 40.85% | 26.41% | 85.54% | 54.42% | 61.81% | 48.60% | 61.85% |
| HunyuanCustom [35] | Open-Source | 56.89% | 44.84% | 17.94% | 86.49% | 55.93% | 62.71% | **56.49%** | 58.98% |
| MAGREF-480P [19] | Open-Source | 53.44% | 46.31% | 27.43% | 92.63% | 33.77% | 69.02% | 42.45% | 68.33% |

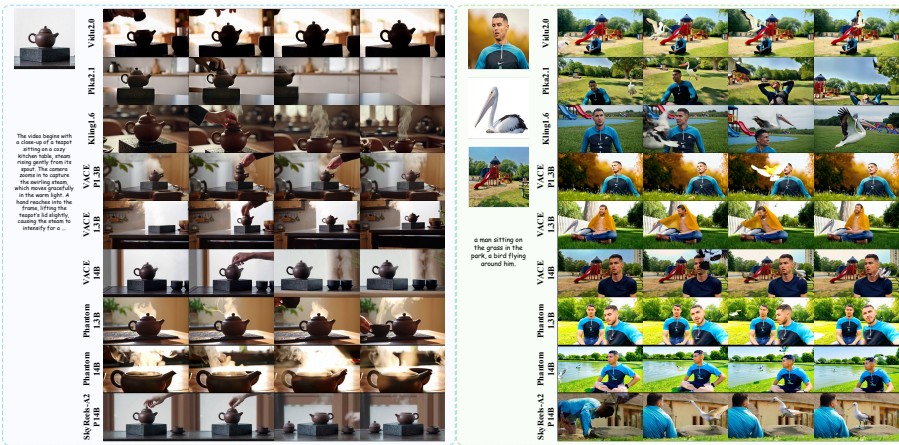

Figure 6: **Qualitative Comparison among Different Methods for the Open-Domain Subject-to-Video task.** Existing methods handle non-human entities better than human identities, and perform better with single subject compared to multiple subjects.

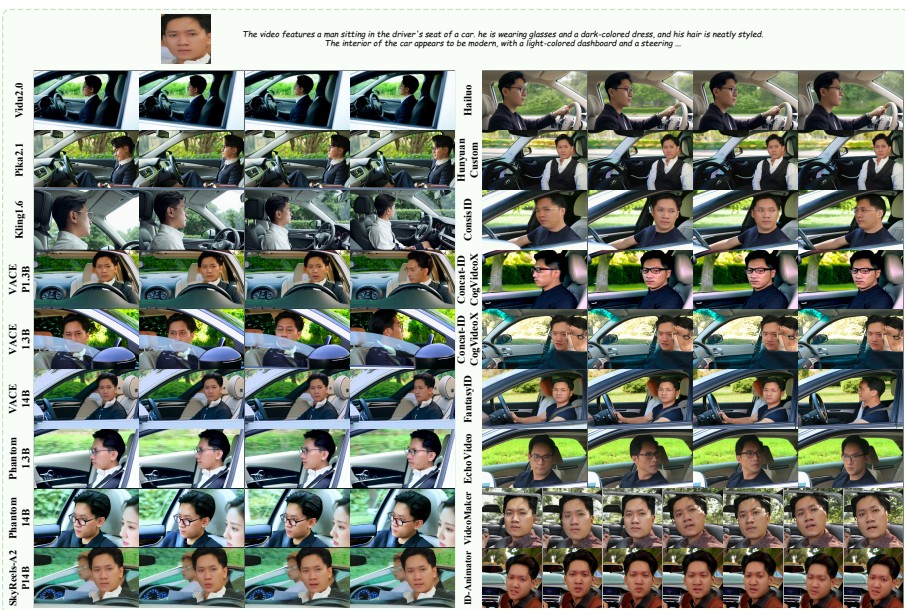

Figure 7: **Qualitative Comparison among Different Methods for the Human-Domain Subject-to-Video task.** They are unable to generate consistent side profiles and suffer from copy-paste issues.

and GmeScore—align with human perception and accurately reflect the subject consistency, subject

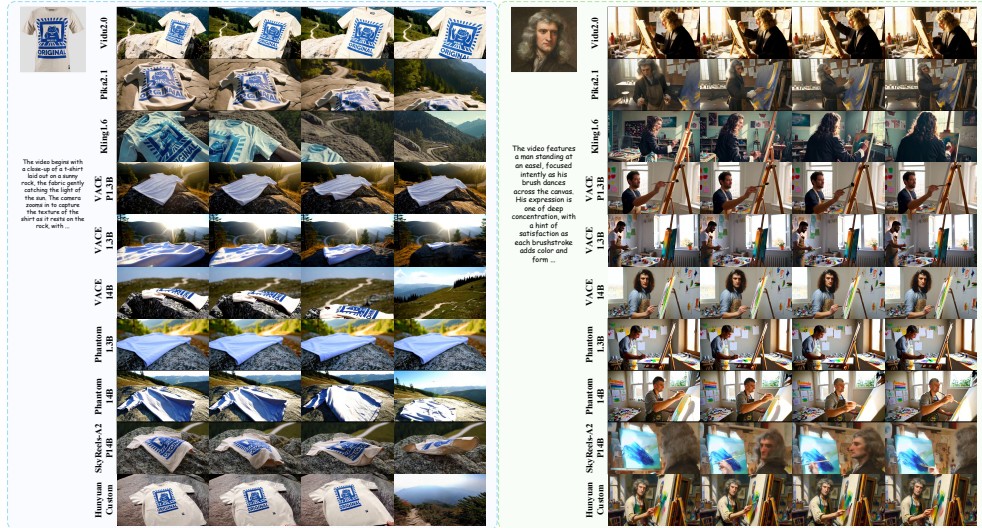

Figure 8: **Qualitative Comparison among Different Methods for the Single-Domain Subject-to-Video task.** Existing models perform better on single-subject than multi-subject tasks.

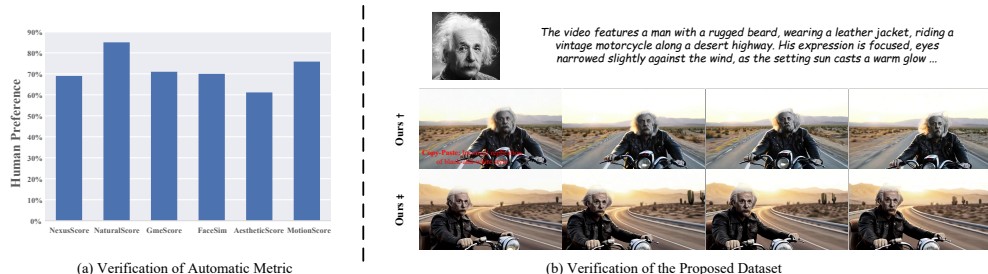

(a) Verification of Automatic Metric

(b) Verification of the Proposed Dataset

Figure 9: **(a) Alignment between Automatic Metrics and Human Perception.** The proposed metrics are comparable to other metrics [17, 6, 16] in terms of human preference. **(2) Validation of ConsisID-Nexu-5M with † and without ‡ Nexus Data.** Training are based on ConsisID [119].

naturalness, and text relevance. Moreover, the proposed metrics are comparable to other metrics [17, 6, 16] in terms of human preference. Further details can be found in the Appendix D.6.

**Validation of OpenS2V-5M.** Finally, to evaluate the effectiveness and robustness of OpenS2V-5M, we fine-tune a model initialized with Wan2.1 1.3B weights [92] using the ConsisID method [119], employing only MSE loss and omitting mask loss. Given computational constraints, we randomly use 300k samples from OpenS2V-5M, focusing solely on single human identity during training. The results, presented in Figure 9(b) and Table 7, demonstrate that our dataset successfully converts a text-to-video model into a subject-to-video model, thus validating the proposed dataset and its data collection pipeline, especially the Nexus Data plays a crucial role. Since the model is not fully trained, it has not yet achieved optimal performance and is intended for verification purposes only.

## 6 Conclusion

In this paper, we present OpenS2V-Eval, the first benchmark specifically designed for evaluating subject-to-video (S2V) generation. This benchmark addresses the limitations of existing benchmarks, which are primarily derived from text-to-video models and overlook crucial aspects such as subject consistency and subject naturalness. Additionally, we present three new automated metrics aligned with humans—NexusScore, NaturalScore, and GmeScore. Furthermore, we introduce OpenS2V-5M, the first open-source million-scale S2V dataset, which not only includes regular subject-text-video triples but also incorporates Nexus Data constructed using GPT-Image-1 and cross-video associations, thus promoting further research within the community and resolving the three core issues of S2V.

# 7  Acknowledgments

We thank all the anonymous reviewers for their constructive comments. This work was supported in part by the Natural Science Foundation of China (No. 62332002, 62202014, 62425101).

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

# Paper Appendix for *OpenS2V-Nexus*: A Ultra-Scale Dataset and Benchmark for Subject-Consistent Video Generation

## A  Related Works: Subject-Consistency Video Generation Models

Diffusion models are widely acknowledged for their remarkable generative capabilities [78, 77, 75, 66, 67, 65, 87, 118, 24], which have significantly advanced the development of subject-consistency generation models [40, 29, 28, 10]. Initially, researchers utilized tuning-based methods to generate consistent image content, such as DreamBooth [80], Lora [32], and Textual Inversion [25]. These methods integrate specific reference content into the training process through fine-tuning existing parameters, adding extra parameters, or modifying text embeddings. Later models, including Magic-Me [68], MotionBooth [106], and DreamVideo [101], extended these approaches to video generation. However, since these methods require training on each new reference content before inference, their practical application is limited. To mitigate the high computational cost, tuning-free methods were

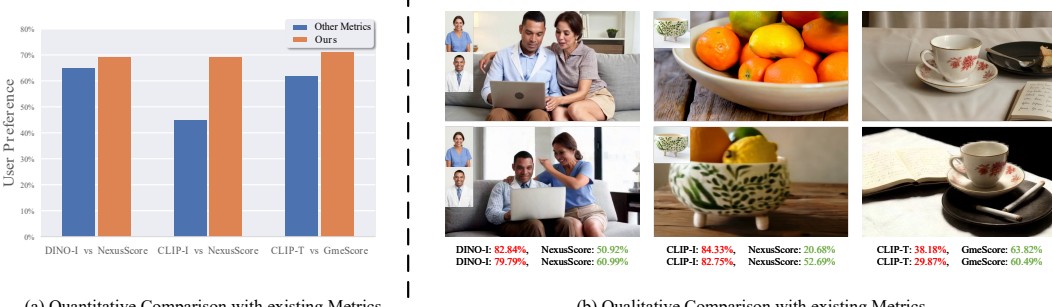

(a) Quantitative Comparison with existing Metrics          (b) Qualitative Comparison with existing Metrics

Figure 10: **Comparison with Existing Metircs for Subject Consistency and Text Relevance.** The proposed automatic metricsalign more closely with human preferences compared to the commonly used DINO-I [122], CLIP-I [76], and CLIP-T [76] in existing S2V methods [42, 58, 39, 22].

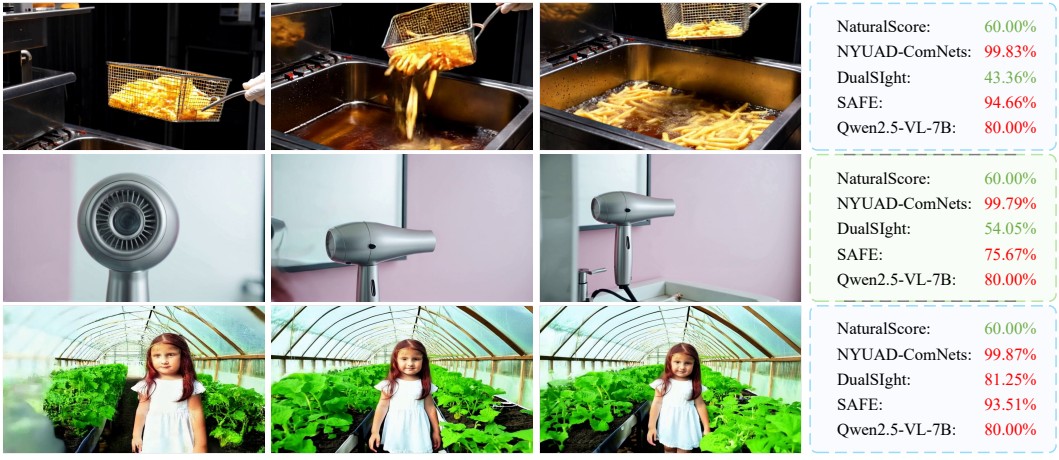

Figure 11: **Comparison with Existing Methods for Subject Naturalness.** Existing AIGC anomaly detection models and multimodal models are both prone to misidentifying generated content as real.

introduced. A notable example is IP-Adapter [114], which leverages large datasets to train additional adapters for open-domain subject-consistency generation. However, due to its lower fidelity to human identity, InstantID [95] and PhotoMaker [49] developed human-domain subject-consistency generation models based on this approach. Similar to these image consistency techniques, ID-Animator [31] and ConsisID [119] achieved tuning-free Subject-to-Video (S2V) generation on UNet and DiT, respectively. Nevertheless, these approaches [129, 100, 23, 126] are confined to the human domain, limiting their broader applicability. Recent works, such as Phantom [58], VACE [42], and SkyReels-A2 [22], have demonstrated the ability to generate consistent multi-subject videos in the open domain [51, 13, 37], gradually narrowing the gap with commercial S2V models [45, 46, 90, 5]. However, a unified and comprehensive benchmark to assess the strengths and weaknesses of these models remains absent, and the lack of publicly released training data impedes further progress in this field. Therefore, we introduce OpenS2V-Eval and OpenS2V-5M, aimed at bridging this gap.

## B    More Details of OpenS2V-Eval

### B.1    Comparison with Existing Metircs for Subject Consistency and Text Relevance

As previously noted, Alchemist-Bench [13], VACE-Benchmark [42], and A2 Bench [22] enable the evaluation of open-domain S2V. However, these evaluations are typically derived from VBench [39] and are predominantly limited to global, coarse-grained assessments. Specifically, they often rely on CLIP [76] or DINO [122] to calculate the similarity between text and images, both of which have been shown to exhibit poor robustness [104, 121, 61]. To substantiate these claims, we employ an

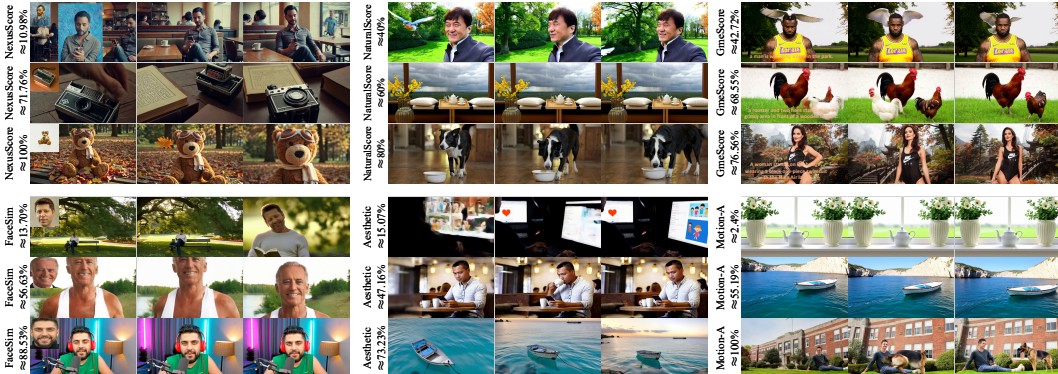

Figure 12: **Visual Reference for Varying Scores of Different Metircs.** It is evident that the proposed NexusScore, NaturalScore, and GmeScore are highly correlated with human perception.

evaluation akin to human evaluation to gather user preferences for DINO-I, CLIP-I, and CLIP-T. Additionally, six samples are randomly selected for qualitative analysis, as illustrated in Figure 10. The results demonstrate that the proposed NexusScore and GmeScore offer greater accuracy in assessing subject consistency and text relevance compared to others. All higher scores are better.

## B.2 Comparison with Existing Metrics for Subject Naturalness

To evaluate whether a generated video is natural—meaning whether it complies with the laws of physics and common sense—a simple solution is to apply AIGC anomaly detection models [111, 48, 69, 2, 71], using the probability of the real label as the score. Alternatively, open-source multimodal large language models [3, 93, 53, 88] can be used for video scoring. However, we found that the former lacks accuracy, while the latter suffers from poor instruction-following performance and is prone to significant hallucinations. None of these methods perform as effectively as the NexusScore we propose, which is based on GPT-4o [1], as shown in Figure 11.

## B.3 Visual Reference of Different Metrics

We also provide visual samples of NexusScore, NaturalScore, GmeScore, FaceSim-Cur [119], AestheticScore [16], and Motion-A [6] with different scoring scales, as shown in Figure 12. It can be observed that all the metrics are consistent with human perception, especially the three proposed automatic metrics targeting subject consistency, subject naturalness, and text relevance.

## B.4 More Qualitative Analysis

We present further qualitative analysis, as illustrated in Figures 13, 22, 21, and 23. Both open-source and closed-source models encounter the following challenges:

**Poor Generalization** Although open-domain S2V models claim to support input from images of any category, they do not consistently produce satisfactory results. As illustrated in case 5 of Figure 21, while Kling [45] largely preserves the mole's body shape, it loses the original fur color. Other models [46, 58, 22] entirely lose the reference subject information. Furthermore, as the number of reference images increases, the model's ability to retain information progressively diminishes. This issue is particularly pronounced in open-source models [22, 42], as shown in cases 1–6 of Figure 21.

**Copy-Paste Issue** Existing models often inaccurately replicate the lighting, pose, expression, and other attributes from reference images directly onto generated videos, instead of generating content by learning the intrinsic features of the reference subjects. Although this may result in higher fidelity content, it generally fails to align with human perception and appears unnatural. As illustrated in Figure 13(c), the model directly places a face onto a person leaning against a pillar, creating an unnatural and visually awkward effect. This problem is particularly evident in generating human.

**Inadequate Human Fidelity** As demonstrated in Figures 21, 23, and 24, current models often face difficulties in preserving human identity as effectively as they preserve non-human entities.

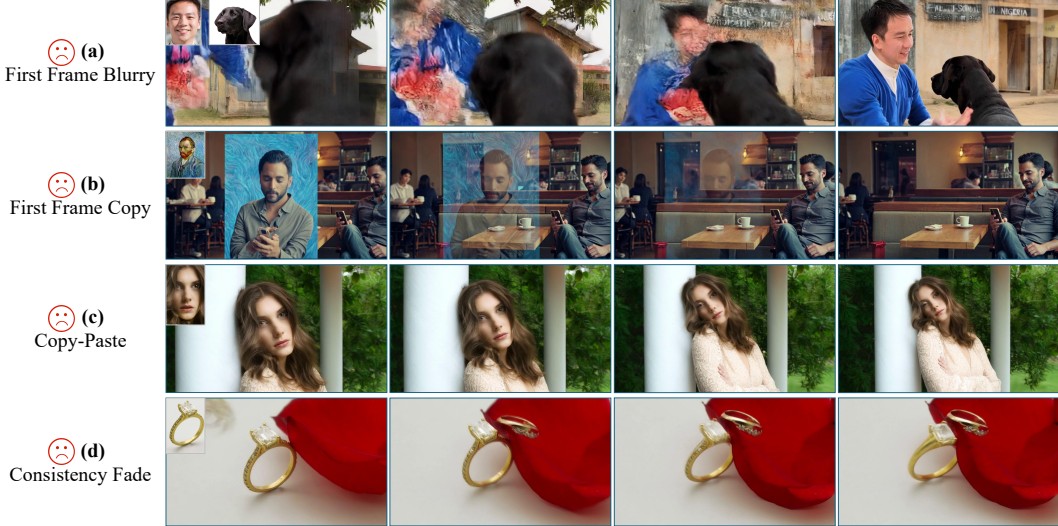

Figure 13: **Example of Common Issues faced by current Subject-to-Video Generation Models.** These videos are generated by Kling [45] and SkyReels-A2 [22] for demonstration purposes only.

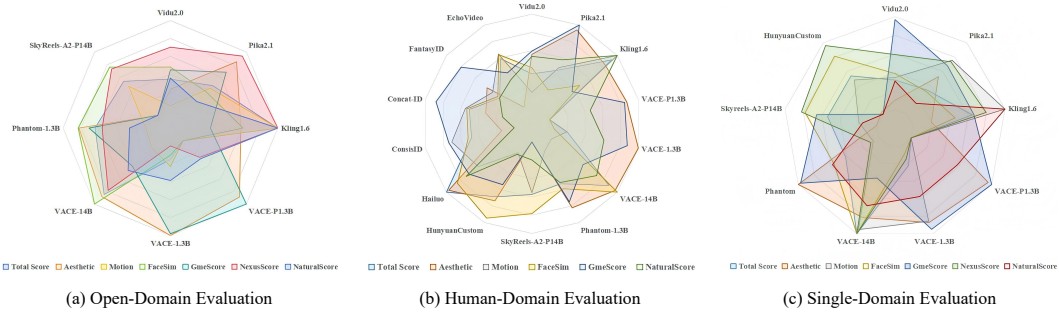

(a) Open-Domain Evaluation    (b) Human-Domain Evaluation    (c) Single-Domain Evaluation

Figure 14: **Visualization of all the Quantitative Results in OpenS2V-Eval.**

While part of this issue can be attributed to human perception being more sensitive to facial changes, the primary cause lies in the models' insufficient capabilities. This is also one of the reasons why human-domain models exist, such as ConsisID [119], EchoVideo [100] and Hailuo [90].

**First Frame Blurry or Copy**    In addition to the three core issues outlined above, we also observe a noteworthy phenomenon in which the model directly replicates the reference image into the generated video, as illustrated in Figure 13(b), generated by Kling [45]. Furthermore, it is possible that the first few frames of the generated video appear blurry, gradually becoming clear as shown in Figure 13(a), generated by SkyReels-A2 [22]. Similar phenomena are also observed in the Phantom [58], ConsisID [119], and Concat-ID [129] models, likely due to the use of VAE [11, 50] as the control signal.

**Consistency Fade**    As shown in Figure 13(d), although the model effectively preserves both global and local information of the subject in the first half of the video, the diamond embedded in the ring gradually disappears as the sequence progresses. This issue may stem from the underlying video generation model [92, 43, 113], but it remains a noteworthy concern.

### B.5    Guideline for Model Selection

We visualize all the results of OpenS2V-Eval, as shown in Figure 14. As the number of S2V models increases, the community faces challenges in selecting the most appropriate model, as each one tends to highlight its best results. To address this challenge, we offer model selection guidelines based on the evaluation outcomes of OpenS2V-Eval: (1) For content creators (e.g., advertisements, product displays), the closed-source Kling [45] is the clear leader, providing a more flexible and user-friendly experience. However, due to its high inference cost, more cost-effective alternatives such as Pika

[46] and Vidu [5] may be preferred. While these alternatives do not surpass Kling [45], they still outperform open-source models. (2) For community developers, it is recommended to base S2V model development on Phantom [58] or VACE [42], as it generates videos with relatively high quality and subject fidelity. Fine-tuning these methods can reduce development costs. (3) Although Hailuo has a narrower scope of application, it outperforms open-domain models like Kling in preserving human identity, making it more suitable for generating human-centric videos, such as those involving models and voice-over content. (4) For developing human-centric S2V models, open-source methods like HunyuanCustom [35], and ConsisID [129] offer high-quality pretrained weights, which may could also be extended to open-domain subject-to-video generation.

## C  More Details of OpenS2V-5M

### C.1  Additional Details of Subject-Driven Processing

**Human-Centric Filtering.**  Our data comes from 14,818,489 raw videos crawled from Internet through the Open-Sora Plan [52], consisting of no transition, clean clips with detailed raw captions. We design 100 human-related verbs and nouns as search terms, which lead to the identification of 12,654,783 human-related videos based on the raw captions. Finally, we apply the Aesthetic Predictor [16], the OpenCV [6], the DOVER [102], and the OCR model [91] to obtain aesthetic scores, motion scores, technical scores, and watermark-free video areas, respectively, and filter out low-quality data, ultimately yielding 5,437,544 high-quality clips.

**Subject-Driven Annotation.**  Unlike text-to-video, subject-to-video data requires captions that emphasize the subject. To achieve this, we first use Qwen2.5-VL-7B [93] to describe the appearance and changes of the subject while preserving essential elements of the video, such as environmental context and camera movements, to get the subject-centric video caption. Next, to obtain high-quality reference images, we use DeepSeekV3 [56] to extract keywords related to the environment and objects from the caption. We then input the first frame of the video and these keywords into GroundingDino [59], an open-vocabulary object detection algorithm, to extract reference images for each video. Finally, the bounding boxes obtained from the previous step are fed into SAM2.1 [79], which generates a mask for each subject. This mask can be used to extract reference images without background pixels. To ensure data quality, we further assign Aesthetic Score [16] and text GmeScore to the reference images, allowing users to adjust thresholds to balance data quantity and quality.

### C.2  Additional Details of Dataset Statistics

OpenS2V-5M is the first high-quality, large-scale S2V dataset. In contrast to standard datasets [47, 9, 12], it includes Nexus Data specifically designed to address three critical challenges faced by S2V methods. As depicted in Figure 15, the word cloud illustrates the dataset's rich visual content. Regarding video duration, the majority (91%) of videos are between 0 and 10 seconds, while the remaining videos exceed 10 seconds. In terms of resolution, 65% are 720P, with the rest being high-resolution videos. The captions primarily consist of detailed descriptions, with a wide range of word usage. These settings are tailored to the emerging DiT-based models [62, 43, 113, 92], which favor long prompts and are constrained by input limitations, such as 81 frames and 480P resolution. Furthermore, low-quality videos were excluded during preprocessing based on motion, technical, and aesthetic scores, ensuring that most videos are of high quality. Due to resource constraints, we select the top 10K samples with the highest average scores from the 5M dataset to construct gpt-frame pairs. For cross-frame pairs, we identify 0.35M clustering centers from the regular data, each containing an average of 10.13 samples, meaning we could theoretically create far more than 0.35M $\times$ 10.13 pairs.

### C.3  Further Verification on OpenS2V-5M

Due to limited space in the main text, we provide additional qualitative analysis of Ours‡ here, with results shown in Figure 16. It can be observed that Ours‡ is capable of generating high-quality videos, thereby validating the effectiveness of the proposed OpenS2V-5M.

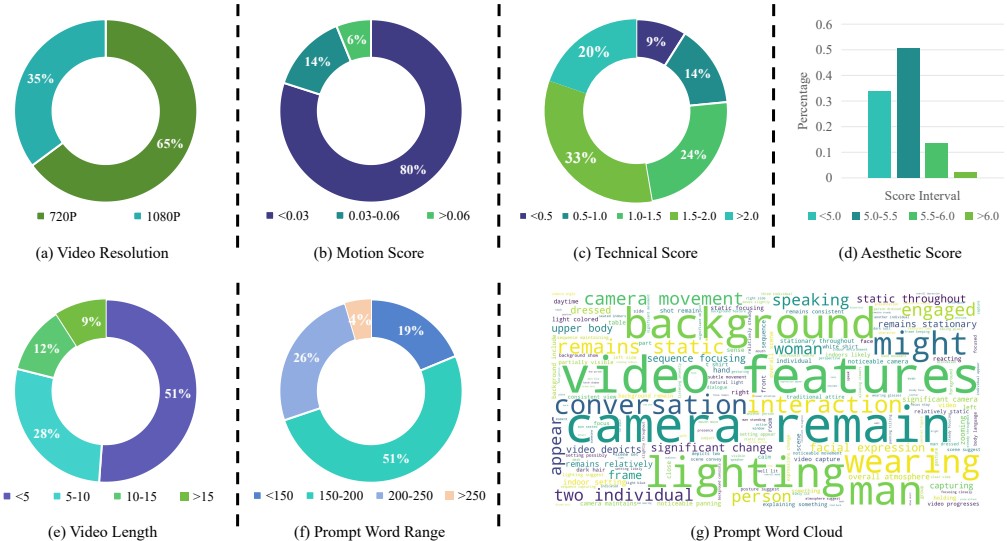

Figure 15: **Statistics in OpenS2V-5M.** The dataset includes a diverse range of categories, clip durations and caption lengths, with most of videos being in high quality (e.g., resolution, aesthetic).

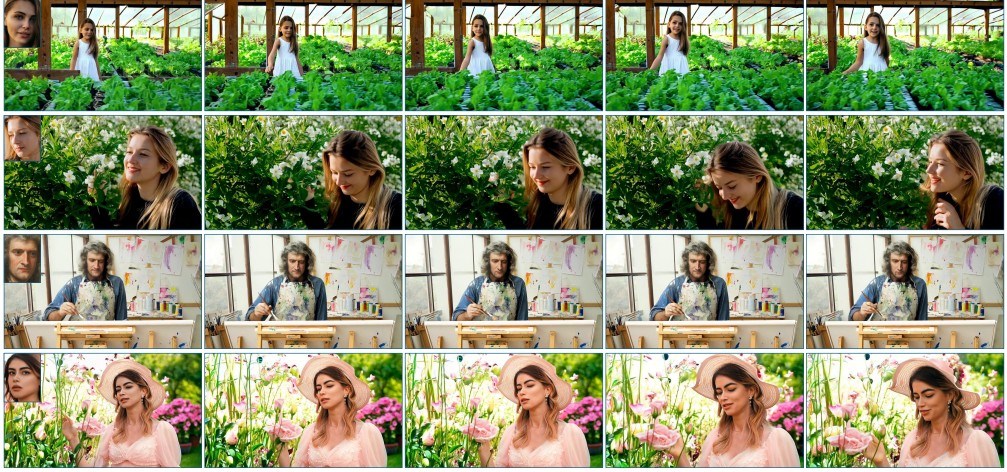

Figure 16: **More Showcases Generated by Ours‡.**

## C.4 Samples of Collected Data

Figure 17 presents diverse samples from the OpenS2V-5M dataset, which consists of subject-text-video triples across multiple categories, offering rich visual information. The subjects include both regular data obtained through segmentation and Nexus Data generated via cross-video association and GPT-Image-1, encompassing humans, objects, backgrounds, and more. These samples highlight the dataset's diversity and depth, and are expected to address the three primary challenges faced by subject-to-video generation models, thereby advancing the field and contributingto the community.

## D More Details of Experiment

### D.1 Details of Resource

We employ Nvidia A100 (x40) for all the experiments. All implementations are conducted on the basis of the official code using the PyTorch framework or official interface.

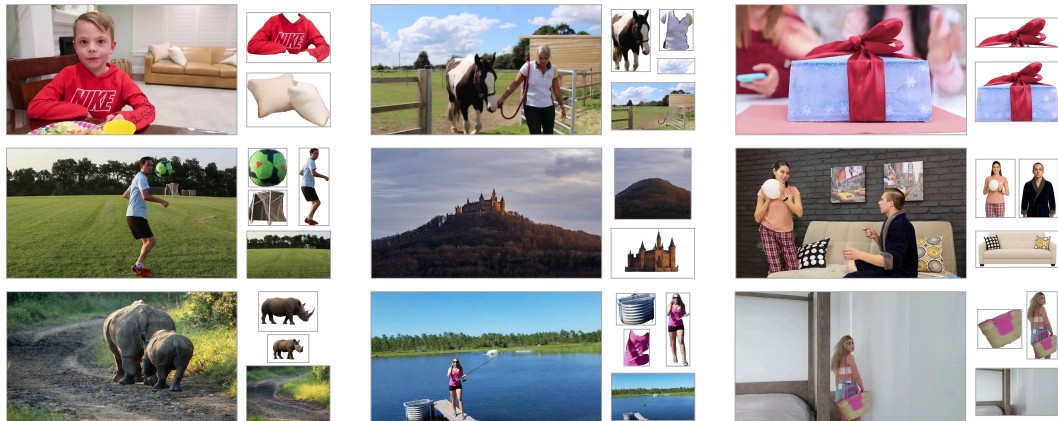

Figure 17: **Samples from the OpenS2V-5M dataset.** The dataset consists of subject-text-video triples, which exhibit more physical knowledge than existing large-scale T2V dataset [12, 94].

## D.2 Details of Evaluation Models

As most S2V models [119, 129, 18, 58, 22, 42] do not support dynamic resolution or variable duration, standardization of these parameters is infeasible. Therefore, we adopt the commonly used official settings [61, 38, 84, 105] to maintain fairness across comparisons.

**Vidu**   *Model Details.* Vidu [5] has released three versions of closed-source models: 1.0, 1.5, and 2.0. Among these, versions 1.5 and 2.0 support multi-reference image input, enabling open-domain subject-to-video generation. However, as the technical report has not been published, specific implementation details remain undisclosed. *Implementation Setups.* We employ the official Vidu 2.0 *charactertovideo* feature with default parameter settings. Using the turbo mode, we generate a 4-second video (65-frames) with a spatial resolution of $704 \times 396$, automatic motion amplitude, and a frame rate of 16 fps.

**Pika**   *Model Details.* Pika [46] has developed five iterations of closed-source model, designated as versions 1.0, 1.5, 2.0, 2.1, and 2.2. Notably, versions 2.0, 2.1 and 2.2 incorporate multi-reference image input capability, enabling open-domain subject-to-video generation. However, due to the absence of an official technical report, the underlying implementation details remain undisclosed. *Implementation Setups.* We employ the official Pika 2.1 *pikaadditions* feature with default parameter settings. The generated video maintains a resolution of $1920 \times 1080$ pixels and a frame rate of 24 fps, with a total duration of 5 seconds (121-frames).

**Kling**   *Model Details.* Kling [45] has released five versions of closed-source model: 1.0, 1.6, and 2.0, among which version 1.6 supports the input of multiple reference images for open-domain subject-to-video generation. However, as no technical report has been released for this version, we are unable to obtain further details. *Implementation Setups.* We employ the official Kling 1.6 *multi\-id* feature with default parameter settings. Using the standard mode, we generate a 5-second video (153-frames) with a spatial resolution of $1280 \times 720$, and a frame rate of 30 fps.

**Hailuo**   *Model Details.* Hailuo [90] has released six versions of closed-source model: I2V-01-Director, I2V-01-live, I2V-01, T2V-01-Director, T2V-01, and S2V-01. Among them, S2V-01 supports the input of multiple reference images to achieve human-domain subject-to-video generation. However, since no technical report has been released for this model, we are unable to obtain further details. *Implementation Setups.* We use the S2V function of the official Hailuo-S2V-01, available at Hailuo-S2V-01, and keep the default settings. We generate a 5-second video (141-frames) with a spatial resolution of $1280 \times 720$ and a frame rate of 25fps.

**VACE**   *Model Details.* VACE [42] is a video generation model based on DiT that integrates various inputs in four data modalities—text, image, video, and mask—and unifies multiple video generation and editing tasks within a single model, including open-domain subject-to-video generation. It releases four model weights: VACE-Wan2.1-1.3B-Preview, VACE-LTX-Video-0.9, Wan2.1-VACE-1.3B, and Wan2.1-VACE-14B. The training data consists of over a million text-to-video samples, which it collects and processes internally. *Implementation Setups.* We use the officially released

VACE code and models, maintaining the original settings. For VACE-Wan2.1-1.3B-Preview and VACE-Wan2.1-1.3B, we generate 5-second (81-frame) videos at a spatial resolution of 832×480 and a frame rate of 16 fps. For VACE-Wan2.1-14B, we generate 5-second (81-frame) videos at a spatial resolution of $1280 \times 720$ and a frame rate of 16 fps.

**Phantom**    *Model Details.* Phantom [58] is a video generation model based on DiT that extracts reference image information using both CLIP and VAE, and employs a windowed attention mechanism to reduce computational overhead, enabling open-domain subject-to-video generation. It includes three model weights: Phantom-Seaweed, Phantom-Wan-1.3B, and Phantom-Wan-14, but only Phantom-Wan-1.3B&14B are publicly released. The training data come from panda70M [12], subject200k [14], OmniGen [108], and internal datasets, totaling over 10 million samples. *Implementation Setups.* We use the officially released Phantom-Wan code and model, maintaining the original settings. We generate 5-second (81-frame) videos at a resolution of $832 \times 480$ and a 16 fps.

**SkyReels-A2**    *Model Details.* SkyReels-A2 [22] is a model fine-tuned based on Wan2.1 [92], employing an approach similar to Phantom. It utilizes a dual-stream architecture to enhance the model's response to reference images and textual prompts, enabling open-domain subject-to-video generation. There are four variants in total: A2-Wan2.1-14B-Preview, A2-Wan2.1-14B, A2-Wan2.1-14B-Pro, and A2-Wan2.1-14B-Infinity, but only A2-Wan2.1-14B-Preview has been open-sourced. The training data comes from 2 million high-quality subject-text-video triples collected internally. *Implementation Setups.* We use the officially released SkyReels-A2-Wan2.1-14B-Preview code and model, maintaining the original settings. Videos are generated with a spatial resolution of $832 \times 480$ and a frame rate of 16 fps, resulting in a duration of 5 seconds (81 frames).

**HunyuanCustom**    *Model Details.* HunyuanCustom [35] is a model fine-tuned based on Hunyuan-Video [35], which achieves open-domain subject-to-video generation by injecting ID information into both the MLLM and the video-driven injection module. In theory, it supports the input of multiple reference images, but currently only the weights supporting Single-Subject have been open-sourced. The training data is processed from internally collected and open-source datasets, but the size of the dataset has not been disclosed. *Implementation Setups.* We use the officially released HunyuanCustom-Single-Subject code, maintaining the original settings. Videos are generated with a spatial resolution of $1280 \times 720$ and a 25 fps, resulting in a duration of 5 seconds (129 frames).

**ConsisID**    *Model Details.* ConsisID [119] is a model fine-tuned based on CogVideoX [113], which achieves human-domain subject-to-video generation by decomposing ID information into high- and low-frequency signals and injecting them into DiT via cross-attention. It only supports the input of a single face image. The training data is processed from internally collected data, with a dataset size of approximately 0.1 million. *Implementation Setups.* We use the officially released ConsisID code and model, maintaining the original settings. Videos are generated with a spatial resolution of $720 \times 480$ and a frame rate of 8 fps, resulting in a duration of 6 seconds (49 frames).

**Concat-ID**    *Model Details.* Concat-ID [129] is a model fine-tuned based on CogVideoX [119] and Wan2.1 [92]. It concatenates image features with video latents along the token dimension, thereby avoiding the issue of blurry initial frames. It only supports input of a single face image. The training data is processed from internally collected data, with a dataset size of approximately 1.3 million. *Implementation Setups.* We use the officially released Concat-ID code and model, maintaining the original settings. For CogVideoX version, videos are generated with a spatial resolution of $720 \times 480$ and a frame rate of 8 fps, resulting in a duration of 6 seconds (49 frames). For Wan-AdaLN version, videos are generated with a spatial resolution of $832 \times 480$ and a frame rate of 16 fps, resulting in a duration of 5 seconds (81 frames).

**FantasyID**    *Model Details.* FantasyID [126] is a model fine-tuned from CogVideoX [113] that facilitates identity-consistent generation by constructing multi-view facial datasets, incorporating 3D geometric priors, and utilizing a layer-aware control signal injection mechanism. The model currently supports only single face image input. Its training data are drawn from ConsisID [119], CelebV-HQ [131], and Open-vid [70], comprising approximately 50,000 samples. *Implementation Setups.* We employ the officially released Fantasy-ID code and model while retaining the original settings. Videos are generated at a spatial resolution of $720 \times 480$ and a frame rate of 8 fps, yielding a duration of 6 seconds (49 frames).

**EchoVideo**    *Model Details.* EchoVideo [100] is a model fine-tuned from CogVideoX [113] that employs the multimodal feature fusion module IITF to achieve identity-preserving video generation

through the integration of textual, visual, and facial identity information. The model supports only a single face image as input. The training data are sourced from internal collections and comprise approximately 3.3 million samples. *Implementation Setups.* We employ the officially released EchoVideo code and model while retaining the original settings. Videos are generated at a spatial resolution of $848 \times 480$ and a frame rate of 16 fps, yielding a duration of 3 seconds (49 frames).

**VideoMaker**    *Model Details.* VideoMaker [107] is a UNet-based model fine-tuned from AnimateDiff [27]. It directly inputs reference images into the video diffusion model and utilizes its intrinsic feature extraction process to achieve subject-to-video generation (e.g., only supports 10 categories of subjects). The training data are sourced from CelebV-Text [131] and VideoBooth [41], comprising approximately 0.1M samples. *Implementation Setups.* We employ the officially released VideoMaker code and model while retaining the original settings. Videos are generated at a spatial resolution of $512 \times 512$ and a frame rate of 8 fps, yielding a duration of 2 seconds (16 frames).

**ID-Animator**    *Model Details.* ID-Animator [31] is a UNet-based model fine-tuned from AnimateDiff [27] that employs FaceAdapter and cross-attention to inject facial information. The model supports only a single face image as input. The training data are sourced from CelebV-Text [131] and comprise approximately 15K samples. *Implementation Setups.* We employ the officially released ID-Animator code and model while retaining the original settings. Videos are generated at a spatial resolution of $512 \times 512$ and a frame rate of 8 fps, yielding a duration of 2 seconds (16 frames).

## D.3    Additional Details of Evaluation Settings

Because some models support only a single subject, while others support multiple subjects, we categorize the evaluation tasks into the following three groups:

**Open-Domain Subject-to-Video**    including ① single-face-to-video, ② single-body-to-video, ③ single-entity-to-video, ④ multi-face-to-video, ⑤ multi-body-to-video, ⑥ multi-entity-to-video, and ⑦ human-entity-to-video.

**Human-Domain Subject-to-Video**    including ① single-face-to-video and ② single-body-to-video. In this context, only the face image is input, without the body image.

**Single-Domain Subject-to-Video**    including ① single-face-to-video, ② single-body-to-video, and ③ single-entity-to-video.

## D.4    Additional Details of Implementations

With the exception of Motion Amplitude and Motion Smoothness, which requires the use of all frames, the other metrics (e.g., NexusScore, NaturalScore, GmeScore, FaceSim, AestheticScore) are calculated by uniformly sampling 32 frames to ensure fairness and minimize overhead. Additionally, due to the differing optimal inference settings for each model, it is not feasible to standardize the resolution of generated videos. **(1)** For Motion Amplitude, we use OpenCV [6] to compute this using the *OpticalFlowFarneback*. **(2)** For Motion Smoothness, we use QAlignVideoScore [55] to compute the motion smoothness about the video. **(2)** For FaceSim, following the approach outlined in ConsisID [129], we first apply insightface [17] to detect the face regions in the video frames and the reference image. We then calculate the similarity between these regions in the curricularface [36] feature space. Finally, we average the sum of all valid scores to obtain the FaceSim for the video. **(3)** For AestheticScore, following the method presented in the improved-aesthetic-predictor [16], we directly input the video frames into the model to obtain scores, then compute the average of all valid scores to obtain the AestheticScore for the video. **(4)** For NexusScore, since we have filtered out low-quality $B_{i,t}$ using $c_{i,t}$ and $s_{i,t}$, high-quality scores may be obtained when only one frame of the video is of high quality while the remaining frames are of lower quality. Therefore, after summing and averaging all valid scores, we divide by $T'$ to mitigate this issue. Here, $T'$ refers to the total number of frames in which an object is detected. In addition, this metric is not used to calculate face similarity to improve robustness, which is why we retain FaceSim. **(5)** For NaturalScore, we use *gpt-4o-2024-11-20* [1] as the base model. For each video, we resize the longer side to 512 pixels and run the model three times, taking the average of these results as the score for the video. **(6)** For GmeScore, since it is based on Qwen2-VL [93], which natively supports dynamic resolution and variable duration, no special processing is necessary.

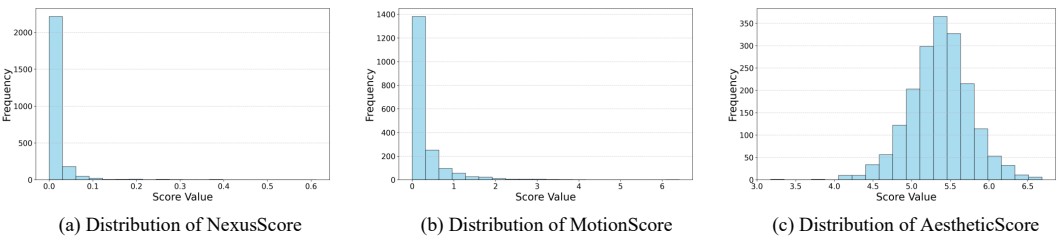

(a) Distribution of NexusScore  (b) Distribution of MotionScore  (c) Distribution of AestheticScore

Figure 18: **Distribution of NexusScore, AestheticsScore and Motion-A.**

### D.5 Additional Details of Metircs Normalization

OpenS2V-Eval evaluates six key dimensions: *subject consistency*, *subject naturalness*, *text relevance*, *face similarity*, *visual quality*, and *motion amplitude*. Due to differing units of measurement across these metrics, direct comparisons and comprehensive analysis are infeasible without normalization. To resolve this, we normalize each metric by defining its theoretical or empirical bounds:

- **FaceSim-Cur**, **GmeScore** and **Motion-S** are bounded by construction, with ranges at $[0, 1]$.
- **NaturalScore** employs a 5-point Likert scale, spanning $[1, 5]$.
- For unbounded metrics (**NexusScore**, **AestheticScore**, and **Motion-A**), we derive ranges of $[0, 0.05]$, $[0, 1]$, and $[4, 7]$, respectively, from their empirical distributions (Figure 18). Out-of-range values are truncated.

To aggregate these normalized metrics into a unified performance score, we compute a weighted sum:

$$\text{Total\_Score} = \sum_{i \in \mathcal{M}} w_i \cdot S_i, \quad \text{where } \mathcal{M} = \{\text{Nexus}, \text{Natural}, \text{Gme}, \text{FaceSim}, \text{Aesthetic}, \text{Motion}\}, \tag{7}$$

with weights $w_i$ assigned as $\iota = 0.20$ (NexusScore), $\kappa = 0.24$ (NaturalScore), $\lambda = 0.12$ (GmeScore), $\mu = 0.20$ (FaceSim-Cur), $\nu = 0.16$ (AestheticScore), $\xi = 0.02$ (Motion-A) and $\sigma = 0.06$ (Motion-S). For humam-domain S2V task, $\kappa = 0.30$, $\lambda = 0.15$, $\mu = 0.25$, $\nu = 0.18$, $\xi = 0.03$ and $\sigma = 0.09$.

### D.6 Additional Details of Human Evaluation

**Pre-processing**   The questionnaire for human evaluation of generated content is developed based on prior studies [119, 121, 78, 84, 83], as shown in Figure 19. The evaluation focuses on six key aspects: *subject consistency*, *subject naturalness*, *text relevance*, *face similarity*, *visual quality*, and *motion amplitude*. For each criterion, a pairwise comparison method is employed, allowing participants to choose between two video options, thereby improving user pleasure and increasing the number of effective questionnaire samples. To ensure category balance, 30 test samples are randomly selected from OpenS2V-Eval, with each sample paired with two videos generated by different models, yielding a total of 60 videos. These videos are annotated with six evaluation scores: NexusScore, NaturalScore, GmeScore, FaceSim-Cur [119], AestheticScore [16], and Motion Quality (Amplitude [6], Smoothness [55]). Taking subject consistency as an example, a sample is labeled as a positive instance for NexusScore if a participant prefers video A over video B and A's NexusScore exceeds that of B; otherwise, it is labeled as a negative instance. The final human preference ratio for each metric is computed as the proportion of positive instances among all test samples. Participants include undergraduate, master's, and doctoral students, as well as members of the general public with no direct affiliation to the research domain. They are drawn from a diverse international pool, including individuals from China, and the United States. This heterogeneous composition ensures both the reliability and generalizability of the evaluation results.

**Post-processing**   Folling [119, 121, 120], to ensure data quality given the use of a five-point evaluation scale, we exclude outlier responses through the following procedures: ① We limit each submission to a single response per IP address and require users to log in prior to voting, thereby ensuring that each participant can submit only one response. ② We assess data validity by considering questionnaire completion time. As it requires 5 to 10 minutes to complete the survey, we exclude responses submitted in less than 5 minutes. ③ We randomize the playback order of videos for each

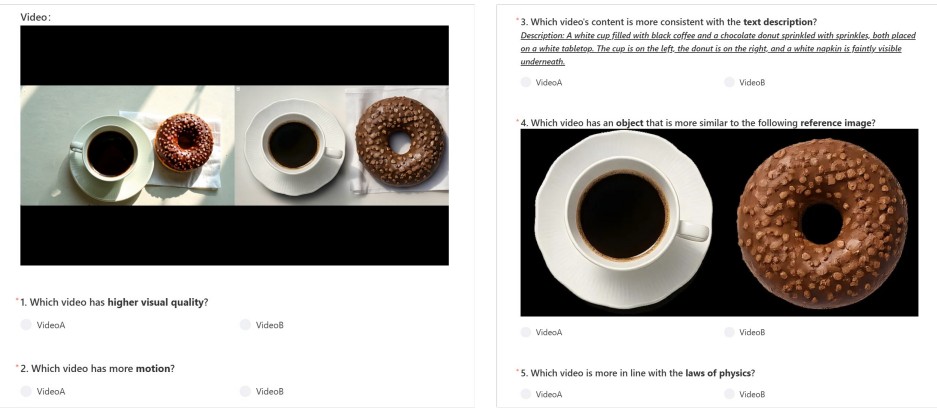

Figure 19: **Visualization of the Questionnaire for User Study.**

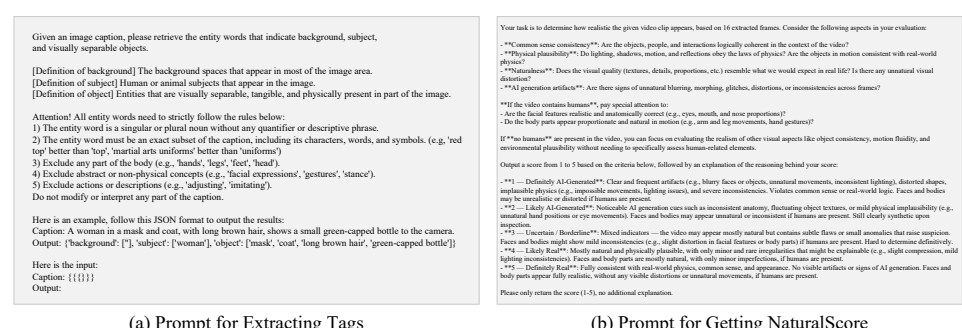

(a) Prompt for Extracting Tags    (b) Prompt for Getting NaturalScore

Figure 20: **Visualization of Different Input Text Prompts.**

participant to mitigate cognitive bias. ④ We implement a sliding verification upon submission to ensure that all questionnaires are completed manually, thereby preventing automated (bot) responses. ⑤ We exclude any questionnaires for which more than 50% of evaluations are extreme values, defined as responses where the sum of the highest (5) and lowest (1) ratings exceeds 50%.

### D.7 Additional Details of Input Prompts

Regarding how to obtain tags through Deepseek [56] and how to annotate videos with NaturalScore using GPT-4o [1], we visualize the input text prompt, as shown in Figure 20.

## E Additional Statement

### E.1 Limitations and Future Work

Although NexusScore and NaturalScore are introduced to evaluate subject consistency and naturalness, these metrics show only approximately 75% correlation with human preferences. Future work aims to better align automated metrics with human judgments. The videos in OpenS2V-5M come from multiple video platforms, and we can only make publicly available those that comply with the CC BY 4.0 license or are copyright-free, totaling approximately 4 million videos.

### E.2 Declaration of LLM Usage

We utilized Large Language Models (LLMs), such as ChatGPT, to support the preparation of this paper. Specifically, LLMs were employed for language-related tasks, including grammar correction, spelling checks, and word choice refinement, to improve the manuscript's clarity and fluency. Additionally, LLMs assisted with data processing and filtering (e.g., our NaturalScore is GPT-based), as well as

generating draft figures to assist the authors in creating refined visualizations. All scientific content, analyses, and conclusions were independently conceived, validated, and interpreted by the authors.

### E.3  Potential Harms Caused by the Research Process

The subject images of **OpenS2V-Eval** are derived from three open-source datasets—ConsisID [119], A2-Bench [22], and DreamBench [74]—that adhere to the Apache license, as well as from three video platforms—Pexels, MixKit, and PixaBay—that operate under the Creative Commons Zero (CC0) license. The video data in **OpenS2V-5M** originates from the Open-Sora Plan [52], with some content licensed under Creative Commons Attribution 4.0 (CC BY 4.0) and others under the Royalty-Free (RF) license. The licensing information for these data is explicitly stated on their respective platforms. The CC0 license designates content as public domain, permitting unrestricted use without additional permissions or authorizations. For CC BY 4.0-licensed videos from the Open-Sora Plan [52], video IDs are included in the metadata to mitigate potential contractual disputes. For RF-licensed videos, we are working to resolve intellectual property issues. In total, approximately 4 million data will be made available as open source. The collected data is organized into seven categories, with contributions from global sources. This diversity ensures that OpenS2V-Eval and OpenS2V-5M are fully representative. The ConsisID model [119] fine-tuned on our dataset demonstrated no significant content bias. Furthermore, video content has been filtered to exclude NSFW material based on subtitle detection. Due to the presence of videos containing identifiable individuals, access to OpenS2V-Nexus is restricted to academic use only, with contact information provided on the https://pku-yuangroup.github.io/OpenS2V-Nexus to ensure the security of personal identity data.

Data collection was made possible through the dedicated efforts of numerous contributors, including the authors of this paper and those involved in the manual evaluation. We consider individual hourly wages or compensation as personal information, and for privacy reasons, these details cannot be disclosed. Nonetheless, we can confirm that all participants have received appropriate compensation in accordance with the legal requirements of their respective countries or regions. The privacy of all participants is safeguarded, ensuring that no additional risks are posed to them.

### E.4  Societal Impact and Potential Harmful Consequences

The objective of **OpenS2V-Eval** is to identify the limitations of existing subject-to-video generation models and to develop the **OpenS2V-5M** dataset to further advance research in this area. While subject-to-video generation models hold significant potential for enhancing creativity, their broader societal impacts must be carefully considered during development:

**First, environmental resource consumption.** Training subject-to-video generation models requires extensive GPU computing power, with a single large-scale training session potentially consuming tens of thousands of kilowatt-hours of electricity, resulting in carbon emissions comparable to the annual emissions of several dozen cars. This high energy consumption not only exacerbates global climate change but also consolidates computational resources within a few dominant tech companies, exacerbating inequality in the research community. To address this, efforts should focus on exploring techniques for model lightweighting, optimizing distributed training efficiency, and promoting the development of green data centers powered by renewable energy to reduce the carbon footprint.

**Second, the risk of linguistic homogeneity and cultural bias.** The text prompt in OpenS2V-Nexus are currently limited to English, which may introduce bias in the model's interpretation of multilingual contexts, such as Chinese. For instance, when generating videos involving non-Western cultural symbols (e.g., Hanfu, Kung fu), the lack of relevant training data could lead to semantic distortions or cultural misinterpretations. Solutions include creating a multilingual annotation system and establishing an open-source collaborative framework to encourage researchers globally to contribute localized data, helping bridge language barriers.

**Finally, the ethical concerns associated with deepfake misuse.** Subject-consistency video generation technologies may be exploited for malicious purposes, such as creating political misinformation, forging celebrity images, or fabricating criminal evidence. The level of realism achievable with these technologies surpasses that of traditional Photoshop techniques. Such misuse poses a threat to public opinion security and judicial integrity. Effective countermeasures should combine technological governance and regulatory oversight: developing generative models embedded with imperceptible watermarks, establishing blockchain-based content traceability protocols, and advocating for legislation

requiring mandatory labeling of generated content. Additionally, public media literacy campaigns should be implemented to enhance society's resilience to false information.

### E.5 Impact Mitigation Measures

We are fully responsible for the authorization, distribution, and maintenance of **OpenS2V-Eval** and **OpenS2V-5M**. Our datasets and benchmarks are released under the CC-BY-4.0 license, while the code is released under the Apache license. We explicitly state on our homepage that all data is intended for academic research purposes to prevent misuse or improper use. We also provide metadata for each video, allowing video creators to contact us promptly and remove invalid videos. All metadata is hosted on *GitHub* and *HuggingFace*, with the following links: https://github.com/PKU-YuanGroup/OpenS2V-Nexus and https://huggingface.co/collections/BestWishYsh.

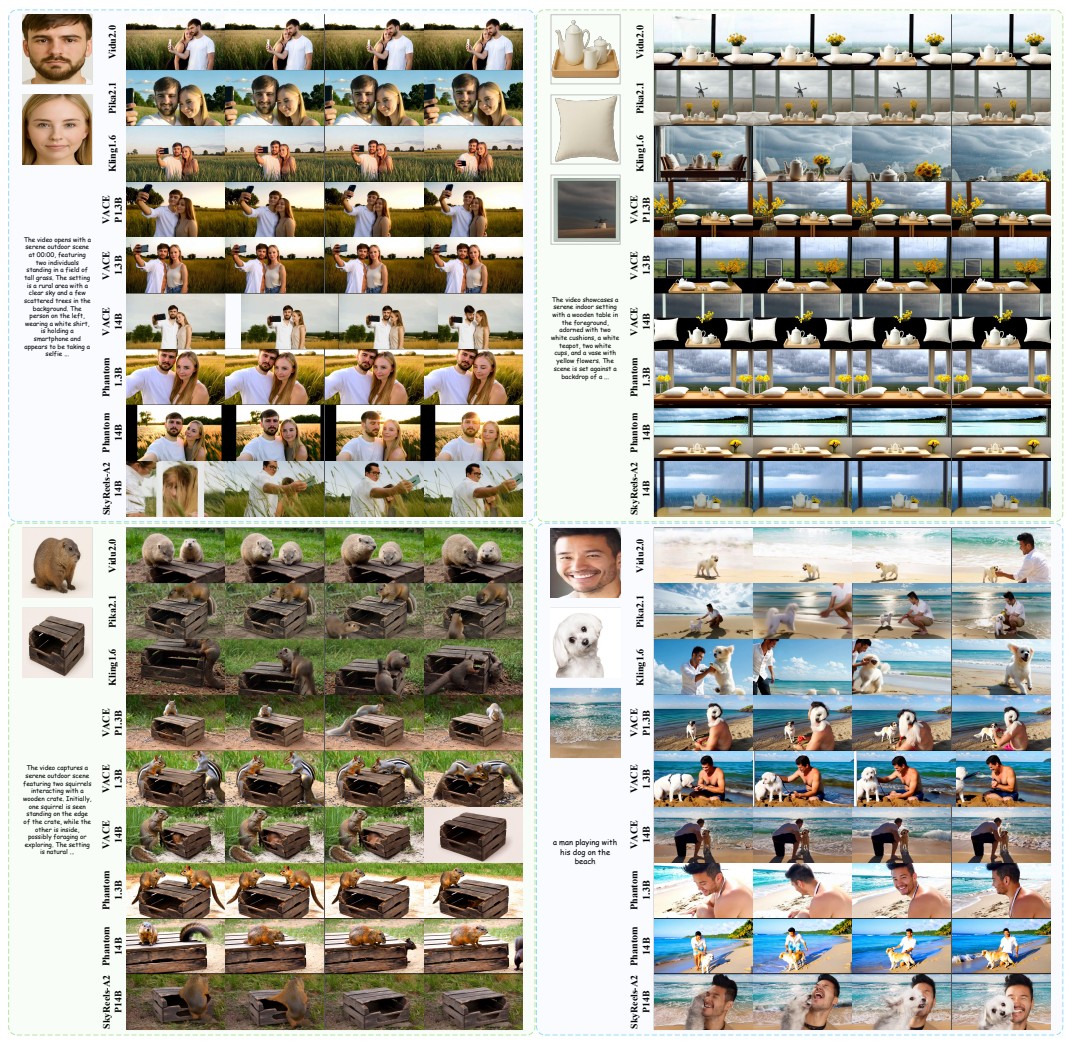

Figure 21: **More Showcases in OpenS2V-Eval for Open-Domain Subject-to-Video Generation.**

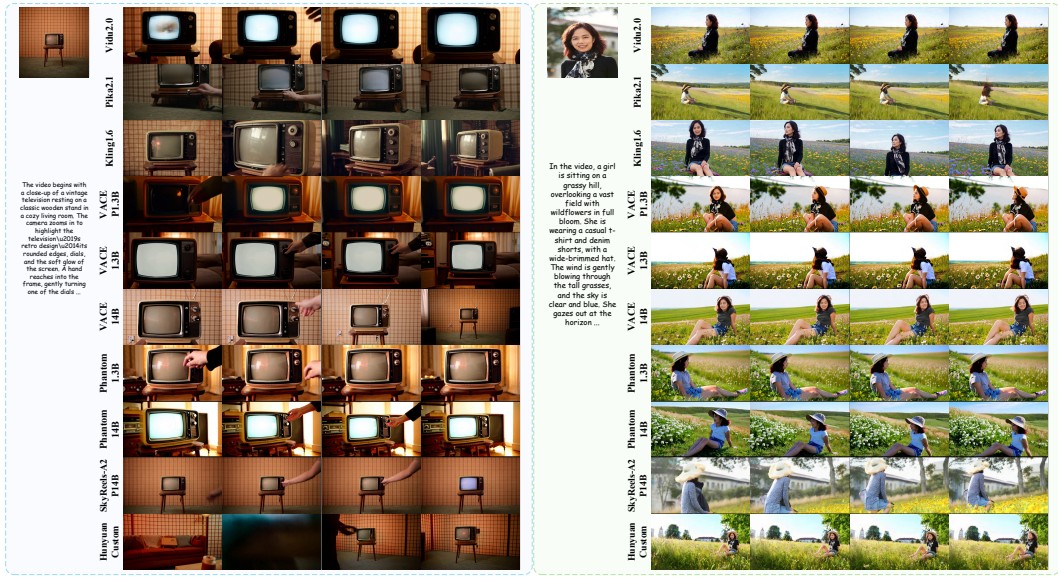

Figure 22: **More Showcases in OpenS2V-Eval for Single-Domain Subject-to-Video Generation.**

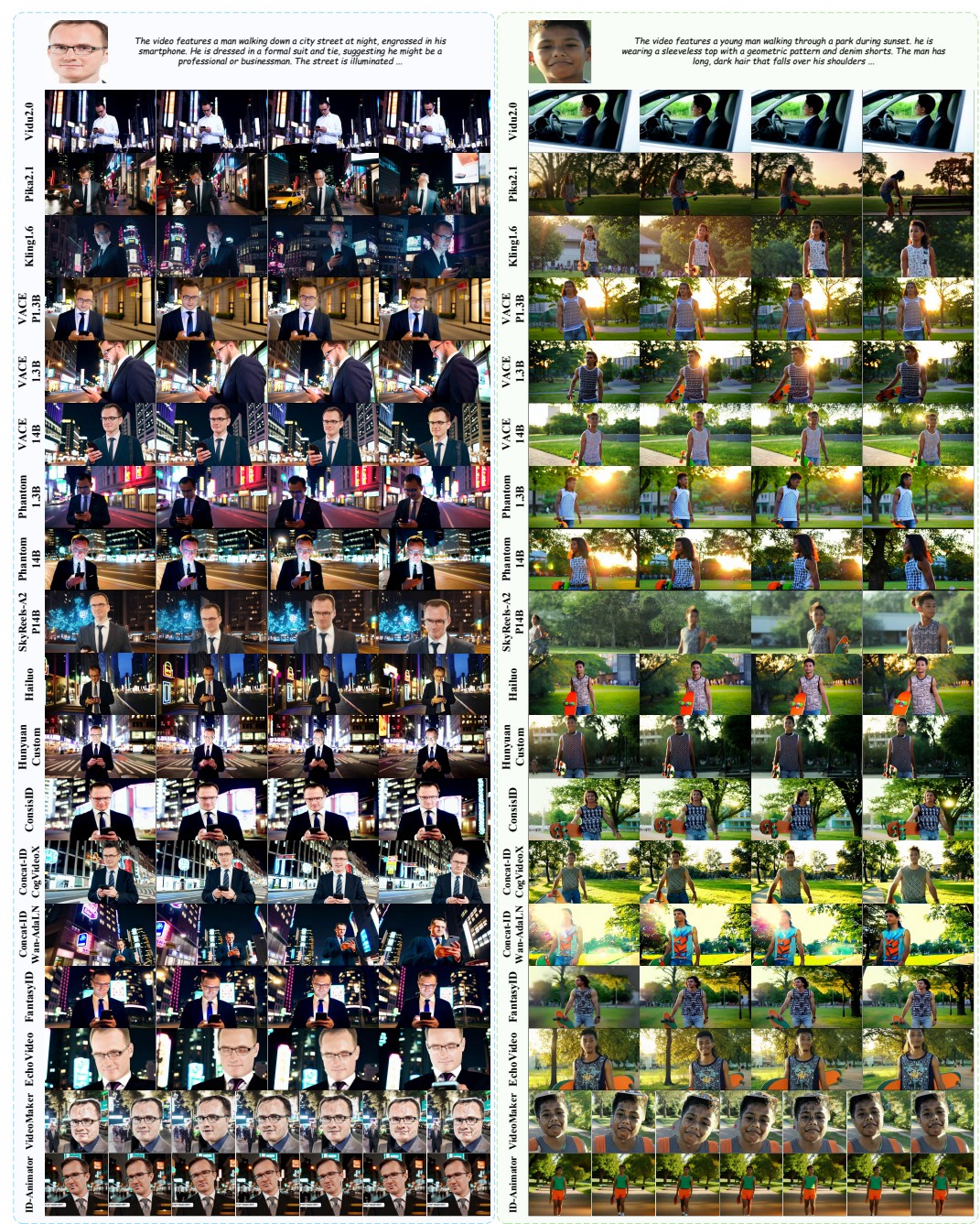

Figure 23: **More Showcases in OpenS2V-Eval for Human-Domain Subject-to-Video Generation.**

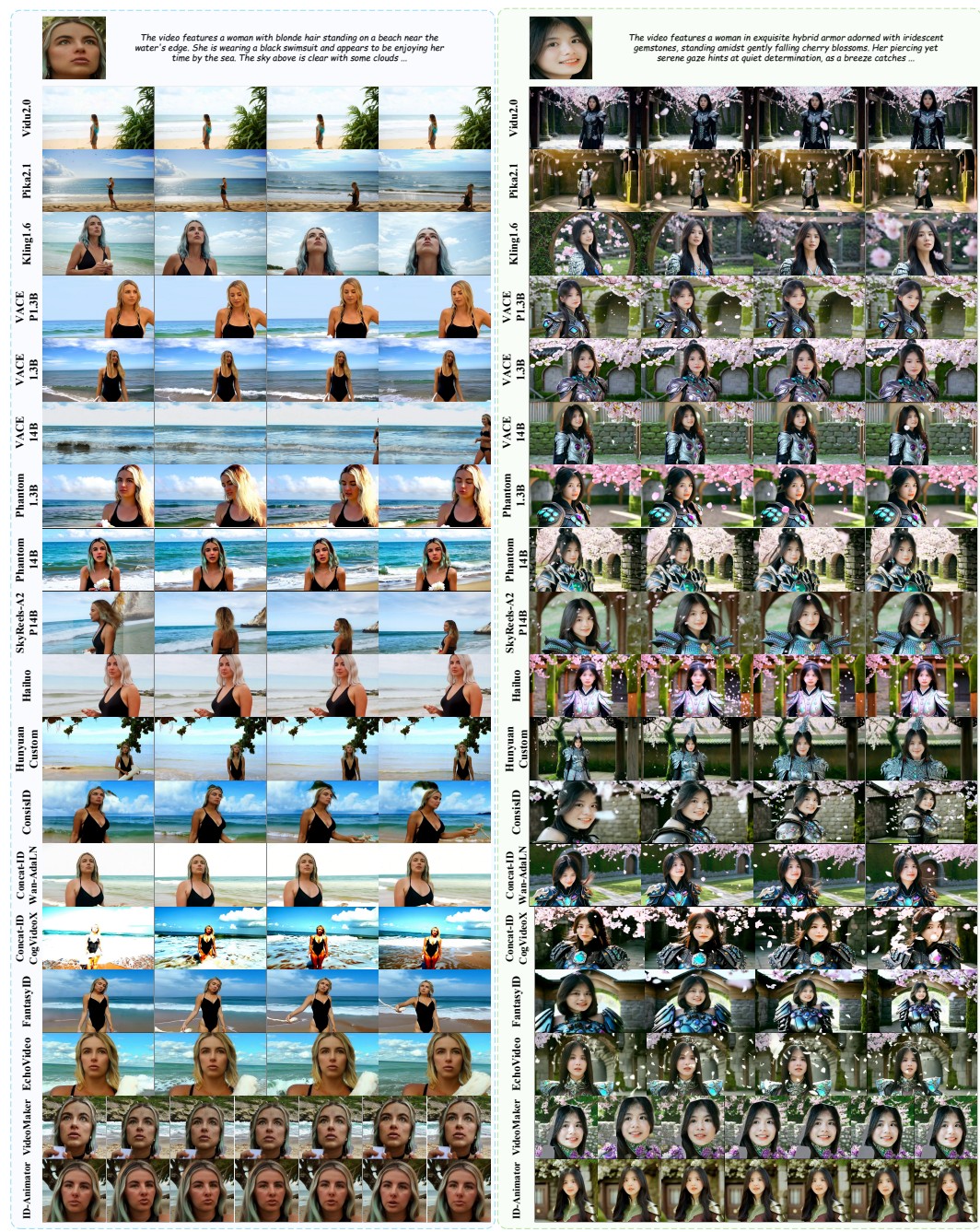

Figure 24: **More Showcases in OpenS2V-Eval for Human-Domain Subject-to-Video Generation.**

