# OpenReview forum: "OpenS2V-Nexus: A Detailed Benchmark and Million-Scale Dataset for Subject-to-Video Generation"
_NeurIPS.cc/2025/Datasets_and_Benchmarks_Track — NeurIPS 2025 Datasets and Benchmarks Track poster_

### Official Review · Reviewer_321W · 2025-06-03

**Rating:** 5
**Confidence:** 4

**Summary:**

The paper introduces OpenS2V-NEXUS, comprising OpenS2V-Eval, a benchmark for S2V generation with 180 prompts across seven categories, and OpenS2V-5M, a dataset of 5.1 million subject-text-video triplets. It proposes three metrics—NexusScore, NaturalScore, and GmeScore—to assess subject consistency, naturalness, and text relevance in video models. An extensive evaluation of various S2V models is conducted to highlight their strengths and weaknesses.

**Dataset Code Accessibility:**

Yes

**Dataset Code Comments:**

By providing the dataset and code, the authors ensure that their findings can be independently verified and replicated by others.

**Ethical Comments:**

The work is focused on advancing the technical aspects of S2V generation without introducing potential harms.

**Ethical Considerations:**

No, there are no or only very minor ethics concerns

**Final Justification:**

Thank you to the authors for addressing my comments. I will maintain my score.

**Limitations Weaknesses:**

- Although the dataset is large, it is unclear if the diversity of subjects, particularly human identity and expression, fully captures real-world contexts. The use of both real and synthetic data, while innovative, poses the challenge of ensuring that synthetic data does not introduce biases or artifacts that could affect model performance in real-world scenarios.

- The paper's reliance on GPT-4o for naturalness scoring and Nexus Data for improved generalization is promising but may limit the model's applicability to real-world data. As primarily a language model, GPT-4o might not fully grasp the nuanced visual dynamics of video generation, especially regarding complex temporal and spatial relationships between objects. This can result in unrealistic or physically inconsistent outputs. Similarly, synthetic data may not adequately challenge the model to learn deeper structures and variations found in natural video content from the real world, potentially leading to models overly optimized for synthetic scenarios.

**Strengths Contributions:**

- OpenS2V-Eval is a substantial step forward in the field, addressing the limitations of existing benchmarks by evaluating subject-to-video generation on multiple key factors, especially focusing on subject consistency and naturalness, which have often been neglected in prior work.
- OpenS2V-5M offers a unique dataset of five million high-quality subject-text-video triplets. The dataset is not only larger than prior S2V datasets but also addresses the core challenges faced by existing S2V models by using both real and synthetic data and leveraging advanced techniques such as GPT-4o for multi-view representations.
- The extensive evaluation of S2V models across different categories of subjects provides an in-depth understanding of the current state of S2V technology, identifying strengths and weaknesses in areas such as fidelity, naturalness, and model generalization.

---

> ### Author Rebuttal · Authors · 2025-07-29
>
> Thank you for the time, thorough comments, and nice suggestions. We are pleased that you acknowledged the metrics, technical soundness, and effective experiments. we are pleased to clarify your questions step-by-step.
>
> >**Q1**: Although the dataset is large, it is unclear if the diversity of subjects, particularly human identity and expression, fully captures real-world contexts. The use of both real and synthetic data, while innovative, poses the challenge of ensuring that synthetic data does not introduce biases or artifacts that could affect model performance in real-world scenarios.
>
> **A1**: We have curated the dataset for breadth from Open-Sora Plan [1]: as detailed in Appendix E.4, it spans seven subject categories with global sourcing, and Appendix C.2 with Fig. 2(g) documents wide category coverage. The CINEMA [2] model fine-tuned on our data shows diverse identities and expressions without obvious collapse (Appendix Fig. 7), indicating that the distribution is adequate for real-world S2V model use.
>
> Specifically, for Nexus (synthetic) data, we extract the subject images as reference images from **real video** frames rather than pure synthetic frames from a text prompt. To this end, we can get a triple data structure (subject image(synthetic), real video, and detailed caption), which is not introduced biases or artifaces for real-world scenarios.
>
> [1] Lin B, Ge Y, Cheng X, et al. Open-sora plan: Open-source large video generation model[J]. arXiv preprint arXiv:2412.00131, 2024.
>
> [2] Deng Y, Guo X, Wang Y, et al. Cinema: Coherent multi-subject video generation via mllm-based guidance. arXiv preprint arXiv:2503.10391, 2025.
>
> >**Q2**: The paper's reliance on GPT-4o for naturalness scoring and Nexus Data for improved generalization is promising but may limit the model's applicability to real-world data. As primarily a language model, GPT-4o might not fully grasp the nuanced visual dynamics of video generation, especially regarding complex temporal and spatial relationships between objects. This can result in unrealistic or physically inconsistent outputs. Similarly, synthetic data may not adequately challenge the model to learn deeper structures and variations found in natural video content from the real world, potentially leading to models overly optimized for synthetic scenarios.
>
> **A2**: **For Data**: OpenS2V-5M contains ~5.4M samples, of which ~5M are real and ~0.35M are Nexus synthetic data (≈93% real).  The former ensures the model’s applicability to real-world scenarios, while the latter improves its generalization capabilities. For Nexus data, we extract the subject images as reference images from **real video** frames rather than pure synthetic images from a text prompt, which keeps the deeper structures and variations found in natural video content from the real world.
>
> **For NaturalScore with GPT-4o**: We acknowledge that GPT-4o may have limited understanding of nuanced visual dynamics of video generation. However, as shown in Appendix B.2 and Figure 2, existing alternatives (e.g., AIGC detectors and other VLMs-based metrics) often fail to distinguish generated content from real. In contrast, Figure 8(a) demonstrates that NaturalScore correlates well with human perception. While not perfect, GPT-4o currently offers the best perceptual alignment among evaluated methods.

---

### Official Review · Reviewer_ja1x · 2025-06-19

**Rating:** 5
**Confidence:** 5

**Summary:**

The paper introduces OPENS2V-NEXUS, comprising:
- OpenS2V-Eval: A benchmark with 180 prompts across 7 S2V categories (e.g., single/multi-entity, human-entity), using real/synthetic data.
- Three new metrics:
    - NexusScore: Measures subject consistency via crop-based detection (addressing background noise).
    - NaturalScore: Evaluates subject naturalness via GPT-4o.
    - GmeScore: Assesses text relevance using MLLMs.
- OpenS2V-5M: A 5.4M 720P video dataset with "Nexus Data" (cross-video associations + GPT-4o-synthesized views) to enhance subject diversity.

**Dataset Code Accessibility:**

Yes

**Ethical Considerations:**

No, there are no or only very minor ethics concerns

**Final Justification:**

All of my concerns have been solved based on the provided rebuttal, especially the updated results on the motion quality analysis. I think this paper can be accepted.

**Limitations Weaknesses:**

1. The concern of the reliability of NaturalScore: Relying on GPT-4o for physical law compliance is problematic—VLMs lack fine-grained physics understanding (e.g., motion dynamics, lighting). This risks misaligning "naturalness" with human perception.

2. Instability of motion metrics: The authors should provide more details on the calculation of motion amplitude since the motion should be as accurate as possible rather than as large as possible. Moreover, I am wondering the stability of the metric since the dynamic degree (similar to motion amplitude) metric in VBench is frustratingly unstable according to my experience.

3. Ambiguous subject count: "No more than 3 subjects per sample" (Sec. 3.2) may limit complex scenarios. A benchmark should push the frontiers of the task by including hard cases.

**Strengths Contributions:**

1. High-impact dataset: OpenS2V-5M is the largest S2V-specific dataset (5.1M regular + 0.3M Nexus Data), resolving data scarcity. Cross-video associations and synthetic multi-view data address generalization issues.

2. Comprehensive benchmark design:
    a) 7 diverse categories (faces, bodies, objects) enable nuanced evaluation.
    b) Inclusion of synthetic data improves testing robustness.

3. Innovative metrics:
    a) NexusScore’s crop-based approach effectively mitigates background noise in consistency evaluation.
    b) Human validation confirms alignment of metrics (Fig. 8a), especially NexusScore/NaturalScore.

4. Comprehensive evaluation:
Tests 11 S2V models (open/closed-source), revealing key insights (e.g., Kling excels in naturalness; Skyreels-A2 suffers "copy-paste" artifacts).

---

> ### Author Rebuttal · Authors · 2025-07-29
>
> Thanks for your time and the constructive suggestions. Your recognition of the paper's high impact and the rigorousness of our research methodologies is greatly appreciated. Here are additional responses and clarifications based on your comments.
>
> >**Q1**: The concern of the reliability of NaturalScore: Relying on GPT-4o for physical law compliance is problematic—VLMs lack fine-grained physics understanding (e.g., motion dynamics, lighting). This risks misaligning "naturalness" with human perception.
>
> **A1**: We acknowledge that GPT-4o may have limited understanding of fine-grained physical laws (e.g., motion dynamics, lighting).
>
> However, as shown in Appendix B.2 and Figure 2, existing alternatives (e.g., AIGC detectors and other VLMs-based metrics) often fail to distinguish generated content from real. In contrast, Figure 8(a) demonstrates that NaturalScore correlates well with human perception. While not perfect, GPT-4o currently offers the best perceptual alignment among evaluated methods.
>
> While current VLMs show limitations in physics understanding, future work could investigate fine-tuning them on physics-aware data to better capture physical plausibility in scoring.
>
> >**Q2**: Instability of motion metrics: The authors should provide more details on the calculation of motion amplitude since the motion should be as accurate as possible rather than as large as possible. Moreover, I am wondering the stability of the metric since the dynamic degree (similar to motion amplitude) metric in VBench is frustratingly unstable according to my experience.
>
> **A2**:
> We have provided the detailed of the motion score in Appendix D.4. As stated, motion amplitude is computed using OpenCV’s OpticalFlow interface, measuring per‑pixel flow magnitudes and aggregating them statistically over frames.
>
> This method is consistent with practices in mainstream pipelines:
>  -  HunyuanVideo[1] filters out static clips by predicting motion speed via estimated optical flow (i.e. use OpenCV's interface),
>  -  Other open‑source models such as Open‑Sora Plan[2], LTX-Video[3] and HuggingFace-video-dataset-scripts also derive motion scores from OpenCV optical flow magnitude.
>
> In comparison, VBench uses the deep model RAFT[4] to estimate optical flow for its “Dynamic Degree” metric. Both routes are valid but imperfect: RAFT is computationally heavier, while OpenCV’s method is lighter, and widely adopted.
>
> At present, our OpenS2V‑Eval prioritizes subject consistency and naturalness over raw motion magnitude, so we adopted the OpenCV‑based motion amplitude as an efficient metric.
>
> [1] Kong W, Tian Q, Zhang Z, et al. HunyuanVideo: A systematic framework for large video generative models. arXiv preprint arXiv:2412.03603, 2024.
>
> [2] Lin B, Ge Y, Cheng X, et al. Open-sora Plan: Open-source large video generation model. arXiv preprint arXiv:2412.00131, 2024.
>
> [3] HaCohen Y, Chiprut N, Brazowski B, et al. LTX-Video: Realtime video latent diffusion. arXiv preprint arXiv:2501.00103, 2024.
>
> [4] Teed Z, Deng J. RAFT: Recurrent all-pairs field transforms for optical flow. ECCV 2020.
>
> >**Q3**: Ambiguous subject count: "No more than 3 subjects per sample" (Sec. 3.2) may limit complex scenarios. A benchmark should push the frontiers of the task by including hard cases.
>
> **A3**: We agree that a benchmark should push the frontier, but the “No more than 3 subjects per sample” cap reflects current S2V models' Iuput/Output limits rather than an arbitrary choice.
>
> For example, current SOTA S2V models (e.g., SkyReels-A2[1], Phantom[2]) support a maximum of 3 images as input. Meanwhile, some baselines (e.g., HunyuanCustom[3], ConsisID[4], VideoMaker[5], and Hailuo[6]) only support a single image as input.
>
> Raising the cap now will disrupt the evaluation process. Empirically, Fig. 6 (main) and Fig. 12 (appendix) already show marked degradation with just two subject images, making three a realistic hard case today.
>
> Our aim is to be stressful yet inclusive. OpenS2V-Eval is designed to scale, and as the multi-subject conditioning model improves, we will add >3-subject splits and report subject-stratified results (1, 2, 3, >3) to continue pushing complexity without sacrificing comparability.
>
> [1] Fei Z, Li D, Qiu D, et al. Skyreels-A2: Compose anything in video diffusion transformers. arXiv preprint arXiv:2504.02436, 2025.
>
> [2] Liu L, Ma T, Li B, et al. Phantom: Subject-consistent video generation via cross-modal alignment. ICCV 2025.
>
> [3] Hu T, Yu Z, Zhou Z, et al. HunyuanCustom: A multimodal-driven architecture for customized video generation. arXiv preprint arXiv:2505.04512, 2025.
>
> [4] Yuan S, Huang J, He X, et al. Identity-preserving text-to-video generation by frequency decomposition. CVPR 2025.
>
> [5] Wu T, Zhang Y, Cun X, et al. Videomaker: Zero-shot customized video generation with the inherent force of video diffusion models. arXiv preprint arXiv:2412.19645, 2024.
>
> [6] Hailuo Video. hailuoai.

---

> > ### Comment · Reviewer_ja1x · 2025-08-03
> >
> > Hi, many thanks for your rebuttal. The answer to Q1 is acceptable and I still encourage you to include splits with more subjects or complex subject attributes, as we believe this would significantly enhance the benchmark's impact within the field. Regrading the motion metric, I understand your answer that you want to emphasize the alignment of your metric to widely used metrics. My idea is that motion amplitude is often a poor proxy for the actual quality of the motion (I think you know it yourself). To create a more robust evaluation, it seems beneficial to explore alternatives. Have you considered other metrics that might better capture critical aspects?

---

> > ### Author Response · Authors · 2025-08-05
> > **Official Comment by Authors**
> >
> > Many thanks for your thoughtful comments and constructive suggestions.
> >
> > **For the number of subject image:**  Based on your suggestion, we have added dev evaluation samples containing 4 and 5 subjects and have updated OpenS2V-Eval on HuggingFace (see the Dataset URLs). In addition, we tested the commercially available models Kling 1.6 and Vidu Q1—which support inputting 4 examples and were released after the submission deadline—and have uploaded the generated videos to HuggingFace as well. We hope this addresses your concern.
> >
> > **For the motion quality:** Our benchmark centers on Subject-to-Video (S2V), where the primary criteria are subject fidelity across frames/views and natural integration under instruction control; motion amplitude is kept as a lightweight activity signal rather than a proxy for motion quality. We agree motion quality should include dynamic stability, dynamic reasonableness, naturalness, and dynamic amplitude. In fact, the first three dimensions have already been measured using the proposed NaturalScore, so we retained only motion amplitude before.
> >
> > For existing unified motion quality metrics, we tested VLM-based (VideoAlign, VMBench) and proprietary-model-based (VBench-Series, ChronoMagic-Bench, FETV-Bench) metrics and found that they were also unstable. Videos with better motion quality could still receive lower scores. To enhance our evaluation framework, we have selected the motion smoothness score from VMBench (*which I think is better than others*) and will incorporate a weighted combination with the motion amplitude score in the camera-ready version (have updated on HuggingFace). We hope this resolves your concern.
> >
> > **Table 1: Human-Domain Subject-to-Video task**
> > |                   |   SkyReels-A2-P14B |   Vidu2.0(20250503) |   EchoVideo |   Hailuo(20250503) |   MAGREF-480P |   Concat-ID-CogVideoX |   HunyuanCustom |   Concat-ID-Wan-AdaLN |   FantasyID |   ConsisID |   Kling1.6(20250503) |   VideoMaker |   VACE-P1.3B |   VACE-1.3B |   Pika2.1(20250503) |   ID-Animator |   Phantom-1.3B |   VACE-14B |   Phantom-14B |
> > |:------------------|-------------------:|--------------------:|------------:|-------------------:|--------------:|----------------------:|----------------:|----------------------:|------------:|-----------:|---------------------:|-------------:|-------------:|------------:|--------------------:|--------------:|---------------:|-----------:|--------------:|
> > | MotionSmoothness↑ |              80.19% |               91.31% |       77.96% |               99.1% |         90.26% |                  81.9% |           84.73% |                 85.86% |       85.44% |      79.83% |                84.75% |         77.5% |         95.8% |       95.84% |               85.29% |         94.69% |          92.02% |      94.96% |         94.81% |
> >
> > **Table 2: Single-Domain Subject-to-Video task**
> > |                   | VACE-P1.3B   | VACE-1.3B   | Kling1.6(20250503)   | Pika2.1(20250503)   | SkyReels-A2-P14B   | MAGREF-480P   | Phantom-14B   | Phantom-1.3B   | VACE-14B   | Vidu2.0(20250503)   | HunyuanCustom   |
> > |:------------------|:-------------|:------------|:---------------------|:--------------------|:-------------------|:--------------|:--------------|:---------------|:-----------|:--------------------|:----------------|
> > | MotionSmoothness↑ | 95.68%       | 95.42%      | 85.76%               | 86.07%              | 85.54%             | 92.63%        | 94.86%        | 93.7%          | 93.16%     | 91.88%              | 86.49%          |
> >
> >
> >
> > **Table 3: Open-Domain Subject-to-Video task**
> > |                   | MAGREF-480P   | VACE-P1.3B   | Phantom-14B   | SkyReels-A2-P14B   | Pika2.1(20250503)   | Kling1.6(20250503)   | Vidu2.0(20250503)   | Phantom-1.3B   | VACE-1.3B   | VACE-14B   |
> > |:------------------|:--------------|:-------------|:--------------|:-------------------|:--------------------|:---------------------|:--------------------|:---------------|:------------|:-----------|
> > | MotionSmoothness↑ | 93.17%        | 96.8%        | 96.31%        | 87.93%             | 87.06%              | 86.93%               | 90.45%              | 93.3%          | 97.2%       | 94.97%     |

---

> > > ### Comment · Reviewer_ja1x · 2025-08-06
> > >
> > > Many thanks for providing the experiments and for analyzing different metrics. I will raise the score because my concern is solved based on the updated results.

---

> > > > ### Author Response · Authors · 2025-08-06
> > > > **Official Comment by Authors**
> > > >
> > > > Thank you for your positive feedback and for increasing your rating. We truly appreciate your thoughtful review and support for our work!

---

### Official Review · Reviewer_XFeP · 2025-06-27

**Rating:** 5
**Confidence:** 5

**Summary:**

The paper considers problems of Subject-to-Video (S2V), different from commonly-considered Text-to-Video and Image-to-Video. It is a novel focus. Specifically, the authors do many pioneering works, such as proposing OpenS2V-Eval and OpenS2V-5M.

OpenS2V-Eval is a fine-grained benchmark, focusing on the model’s ability to generate subject-consistent videos with natural subject appearance and identity fidelity.

OpenS2V-5M is a high-quality dataset that with subject-text-video triplets in 720p resolution.

**Additional Feedback:**

N/A

**Dataset Code Accessibility:**

Yes

**Dataset Code Comments:**

The dataset is available at: https://huggingface.co/datasets/pkuhexianyi/Consisid-Nexus with many downloads.

The code is available at: https://github.com/PKU-YuanGroup/OpenS2V-Nexus with many stars.

**Ethical Considerations:**

No, there are no or only very minor ethics concerns

**Final Justification:**

I appreciate the authors' clarifications in the rebuttal. All of my concerns have been sufficiently addressed. I uphold my recommendation for acceptance and believe OpenS2V can benefit the video generation community.

**Limitations Weaknesses:**

1. The paper refers many times to the Appendix. However, the Appendix is empty. I have checked the supplementary. They are in the Supplementary instead. Please be aware of the difference between Appendix and Supplementary.

2. The paper refers code and data to a research group rather than the specific project page. This behaviour is not professional. The url in DatasetURL is invalid, which is the combination of two URLs. Please do not give more burden to your reviewers.

3. The paper lacks of comparison of SOTA models, like SORA and Veo.

**Strengths Contributions:**

1. The paper does an interesting study on S2V.

2. A comprehensive benchmark is proposed.

3. The writing is easy to follow.

4. The dataset and code are well open-sourced.

5. The paper and proposed datasets and benchmarks will be useful in the video generation community.

---

> ### Author Rebuttal · Authors · 2025-07-29
>
> Thanks for your thorough comments and constructive suggestions. Your endorsement of our benchmark and dataset gives us significant encouragement. Here are our clarifications.
>
> >**Q1**: The paper refers many times to the Appendix. However, the Appendix is empty. I have checked the supplementary. They are in the Supplementary instead. Please be aware of the difference between Appendix and Supplementary.
>
> **A1**:  We sincerely apologize for the confusion caused by the incorrect references to the Appendix (Supplementary). We have moved all of this content to the end of the main text in the revised version.
>
> >**Q2**: The paper refers code and data to a research group rather than the specific project page. This behaviour is not professional. The url in DatasetURL is invalid, which is the combination of two URLs. Please do not give more burden to your reviewers.
>
> **A2**: Since the OpenReview system only allows submitting one URL, we have hosted both OpenS2V-Eval (Data & Leaderboard & Weight) and OpenS2V-5M (Data & Weight) under the separated URL in the combination of the two URLs. We have revised our paper by separating benchmark and dataset URLs and checked the validity to make it clear.
>
>
> >**Q3**: The paper lacks of comparison of SOTA models, like SORA and Veo.
>
> **A3**: *SORA* and *Veo3* does not support the subject-to-video generation, they are mainly focus on text/frame-to-video.
>
> Following your suggestions, we have updated the results of other open-source SOTA models (e.g., *Phantom-14B [1]*, *MAGREF [2]*, and *Concat-ID-Wan-AdaLN [3]*) on the *HuggingFace Leaderboard*, which were released after the submission deadline, and these updates will be included in the camera-ready version.
>
> **Table 1: Open-Domain Subject-to-Video task**
> | **Method**         | **Venue**     | **Total Score$\uparrow$** | **Aesthetics$\uparrow$** | **Motion$\uparrow$** | **FaceSim$\uparrow$** | **GmeScore$\uparrow$** | **NexusScore$\uparrow$** | **NaturalScore$\uparrow$** |
> |--------------------|---------------|---------------------------|--------------------------|----------------------|-----------------------|------------------------|---------------------------|----------------------------|
> | Phantom-14B | Open-Source | 52.32\%                 | 46.39\%                 | 33.42\%              | 51.48\%               | 70.65\%                | 37.43\%                  | 68.66\%                    |
> | MAGREF-480P | Open-Source | 47.93% | 45.02% | 21.81% | 30.83% | 70.47% | 43.04% | 69.49% |
>
> **Table 2: Human-Domain Subject-to-Video task**
> | **Method**                 | **Venue**     | **Domain**      | **Total Score$\uparrow$** | **Aesthetics$\uparrow$** | **Motion$\uparrow$** | **FaceSim$\uparrow$** | **GmeScore$\uparrow$** | **NaturalScore$\uparrow$** |
> |----------------------------|---------------|-----------------|---------------------------|--------------------------|----------------------|-----------------------|------------------------|---------------------------|
> | Phantom-14B | Open-Source   | Open-Domain     | 58.69\%                   | 49.14\%                  | 41.24\%              | 55.02\%       | 72.55\%                | 68.33\%                   |
> | MAGREF-480P | Open-Source | Open-Domain | 50.41% | 51.20% | 14.76% | 32.87% | 70.88% | 72.22% |
> | Concat-ID-Wan-AdaLN | Open-Source   | Human-Domain    | 53.18\%                   | 43.13\%                  | 17.19\%              | 50.05\%               | 71.90\%                | 69.44\%                   |
>
> **Table 3: Single-Domain Subject-to-Video task**
> | **Method**                 | **Venue**     | **Total Score$\uparrow$** | **Aesthetics$\uparrow$** | **Motion$\uparrow$** | **FaceSim$\uparrow$** | **GmeScore$\uparrow$** | **NexusScore$\uparrow$** | **NaturalScore$\uparrow$** |
> |----------------------------|---------------|---------------------------|--------------------------|----------------------|-----------------------|------------------------|---------------------------|----------------------------|
> | Phantom-14B | Open-Source   | 53.17\%                   | 47.46\%          | 41.55\%      | 51.82\%               | 70.07\%       | 35.35\%                  | 69.35\%                    |
> | MAGREF-480P | Open-Source | 49.13% | 46.31% | 27.43% | 33.77% | 69.02% | 42.45% | 69.81%
>
> [1] Liu L, Ma T, Li B, et al. Phantom: Subject-consistent video generation via cross-modal alignment. ICCV 2025.
>
> [2] Deng Y, Guo X, Yin Y, et al. MAGREF: Masked Guidance for Any-Reference Video Generation. arXiv preprint arXiv:2505.23742, 2025.
>
> [3] Zhong Y, Yang Z, Teng J, et al. Concat-ID: Towards Universal Identity-Preserving Video Synthesis. arXiv preprint arXiv:2503.14151, 2025.

---

> ### Comment · Reviewer_XFeP · 2025-08-01
>
> I appreciate the authors' clarifications in the rebuttal. All of my concerns have been sufficiently addressed. I uphold my recommendation for acceptance and believe OpenS2V can benefit the video generation community.

---

> ### Author Response · Authors · 2025-08-01
>
> Dear Reviewer XFeP,
>
> Thanks so much for your kind words and support. We really appreciate it! We’re glad our response addressed your concerns, and we’ll definitely carry forward your goodwill in future work and reviews.
>
> Hope to contribute more to the visual generation community together!
>
> Best,
>
> On behalf of all OpenS2V authors

---

> ### Comment · Reviewer_XFeP · 2025-08-01
>
> By the way, I intentionally included this final justification as an official comment, hoping the authors would see it. It seems that the final justification is not visible to the authors.

---

> > ### Author Response · Authors · 2025-08-01
> >
> > Thanks again for your kind words and for taking the extra step to make your support visible.
> >
> > — We saw it and truly appreciate it! 💗

---

### Official Review · Reviewer_yfg5 · 2025-07-20

**Rating:** 4
**Confidence:** 3

**Summary:**

This paper presents OPENS2V-NEXUS, a comprehensive infrastructure for evaluating and training subject-to-video (S2V) generation models. It includes two major components: (1) OpenS2V-Eval, a benchmark that covers seven categories of S2V tasks with 180 test cases, and three newly proposed automatic metrics for evaluating subject consistency (NexusScore), subject naturalness (NaturalScore), and text relevance (GmeScore); and (2) OpenS2V-5M, the first million-scale S2V dataset, which combines regular subject-text-video triplets and Nexus Data synthesized from cross-video associations and GPT-generated views. The paper conducts extensive experiments on 11 representative S2V models and demonstrates the advantages of the proposed benchmark and dataset in evaluating and improving S2V systems.

**Dataset Code Accessibility:**

Yes

**Ethical Considerations:**

No, there are no or only very minor ethics concerns

**Final Justification:**

I read the rebuttal and the authors addressed most concerns, so I keep the score.

**Limitations Weaknesses:**

* The paper does not show any example videos with specific NaturalScore values (e.g., what a score of 2 vs 5 looks like) or provide the reasoning output from GPT-4o. Without qualitative illustrations and scoring rationales.

* NexusScore does not clarify how failure cases are handled. When no subject is detected in any frame (i.e., T′ = 0), it is unclear whether a zero, NaN, or ignored value is assigned. This impacts the robustness and fairness of the evaluation.

* No ablation or side-by-side comparison is conducted between the new metrics and prior metrics such as CLIPScore, BLIPScore, or AIGC-based anomaly detectors. As a result, it is difficult to quantify the relative gain of the proposed metrics.

**Strengths Contributions:**

* The benchmark is well-designed and covers a broad range of S2V use cases with both real and synthetic data, providing a more fine-grained evaluation than existing alternatives such as VBench or A2 Bench.

* The proposed metrics address known limitations of prior evaluation approaches. NexusScore improves subject localization using a two-stage detection and retrieval pipeline. NaturalScore introduces a GPT-4o-based judgment of visual realism.

* Thorough in experimental evaluation.

---

> ### Author Rebuttal · Authors · 2025-07-29
>
> Thank you for the time, thorough comments, and nice suggestions. Your endorsement of our designs and experiments gives us significant encouragement. We are pleased to clarify your questions step-by-step.
>
> >**Q1**: The paper does not show any example videos with specific NaturalScore values (e.g., what a score of 2 vs 5 looks like) or provide the reasoning output from GPT-4o. Without qualitative illustrations and scoring rationales.
>
> **A1**: Figure 2 and Figure 3 in the supplementary material presents example videos corresponding to different levels of NaturalScore (e.g., 40%, 60%, and 80%). In addition, Figure 11(b)  in the supplementary material provides the rationale behind these scores as generated by GPT-4o. As shown, the NaturalScore exhibits strong alignment with human perception, which higher scores consistently correspond to qualitatively more natural and coherent video outputs.
>
> Furthermore, we have also provided qualitative illustrations and scoring rationales of the proposed GmeScore, NexusScore in the supplementary material.
>
> >**Q2**: NexusScore does not clarify how failure cases are handled. When no subject is detected in any frame (i.e., T′ = 0), it is unclear whether a zero, NaN, or ignored value is assigned. This impacts the robustness and fairness of the evaluation.
>
> **A2**: In practice, we assign a NexusScore of **zero** when no subject is detected in any frame, which ensures our metric's robustness and fairness. To make it clear, we have included this description in the revised version.
>
> >**Q3**: No ablation or side-by-side comparison is conducted between the new metrics and prior metrics such as CLIPScore, BLIPScore, or AIGC-based anomaly detectors. As a result, it is difficult to quantify the relative gain of the proposed metrics.
>
> **A3**: Figures 1 and 2 in the supplementary material (appendix) have presented *quantitative and qualitative ablation studies between the proposed and existing metrics*.
>
> As shown, the proposed automatic metrics exhibit a closer alignment with human preferences than prior metrics, such as those based on *DINO*, *CLIP*, *AIGC anomaly detection*, and *multimodal models* approaches.

---

> > ### Comment · Reviewer_yfg5 · 2025-08-07
> >
> > I appreciate your detailed response. It has satisfactorily resolved my concern.

---

> > > ### Author Response · Authors · 2025-08-07
> > > **Official Comment by Authors**
> > >
> > > We sincerely thank you again for your review and are happy to see that the concerns you raised have been resolved.

---

### Note · Authors · 2025-08-12

We would like to express our sincere gratitude to all reviewers for their insightful comments and constructive feedback, and **are pleased that their concerns have been addressed**. The reviewers unanimously recognized our work's groundbreaking contributions to advancing S2V research (yfg5, XFeP, ja1x, 321W), particularly highlighting: (1) the novelty of our proposed metrics (yfg5, ja1x, 321W); (2) the comprehensive design of OpenS2V-Eval (yfg5, XFeP, ja1x, 321W); and (3) the importance of OpenS2V-5M in addressing data scarcity and improving generalization (yfg5, XFeP, ja1x, 321W). The clarity and organization of the paper were also praised (XFeP).

Based on these , we conclude some noteworthy replies for the reviewers, ACs and PCs:

- **[Reviewer yfg5]** We have included example videos at the submission stage to illustrate different score levels across various metrics. (Figure 3 in supplementary)

- **[Reviewer yfg5]** We have explained how NexusScore handles failure cases and included the updates in the revised version.

- **[Reviewer yfg5]** We have already conducted a side-by-side comparison between the new metrics and previous ones during the submission stage. (Figure 1 and Figure 2 in supplementary)

- **[Reviewer XFeP]** We will properly move all supplementary content to the end of the main paper to improve readability

- **[Reviewer XFeP]** We have explained that the reason for combining two URLs into one was due to system limitations.

- **[Reviewer XFeP]** We have added evaluation results for more SOTA models released after the submission deadline and have kindly clarified that Sora does not support the S2V task.

- **[Reviewer ja1x]** We have described the calculation for motion amplitude (Appendix D.4) and clarified that this work focuses more on subject consistency and naturalness. We also added an evaluation of motion smoothnees.

- **[Reviewer ja1x]** We have explained that the reason for limiting each sample to no more than three subjects is to accommodate the capabilities of existing models, and increase the number to 5 and evaluate kiling1.6, vidu2.0, and viduQ1.

- **[Reviewer ja1x, 321W]** We have clarified that while GPT-4o still falls short of human-level performance, it is currently the best available approach for computing NaturalScore and constructing synthetic data.

- **[Reviewer 321W]** We have clarified the diversity of the dataset and explained how we address the challenges introduced by using synthetic data during training.

---

### Decision · Program_Chairs · 2025-09-18

**Decision:**

Accept (poster)

**Comment:**

The paper proposes OpenS2V-Nexus, a novel dataset for Subject-to-Video generation. It consists of a fine‑grained benchmark (OpenS2V‑Eval) with novel evaluation metrics, and a million‑scale training dataset (OpenS2V-5M), which consists of five million high-quality 720P subject-text-video triplets.

The paper received 2 Borderline Accept and 2 Accept scores pre-rebuttal. The rebuttal addressed the reviewers' concerns, and one reviewer has increased the score from Borderline Accept to Accept. Overall, the paper is applauded by its good writing, a comprehensive benchmark, innovative metrics, a high-impact large-scale S2V-specific dataset, and a comprehensive evaluation.

The ACs checked and agreed on paper acceptance.